# Recent warming trends of the Greenland ice sheet documented by historical firn and ice temperature observations and machine learning

Baptiste Vandecrux[1], Robert S. Fausto[1], Jason E. Box[1], Federico Covi[2], Regine Hock[2, 3], Asa K. Rennermalm[4], Achim Heilig[5], Jakob Abermann[6], Dirk van As[1], Elisa Bjerre[7], Xavier Fettweis[8], Paul C.J.P. Smeets[9], Peter Kuipers Munneke[9], Michiel R. van den Broeke[9], Max Brils[9], Peter L. Langen[10], Ruth Mottram[11], Andreas P. Ahlstrøm[1]

[1]Geological Survey of Denmark and Greenland (GEUS), Department of Glaciology and Climate, Copenhagen, Denmark
[2]Geophysical Institute, University of Alaska Fairbanks, Fairbanks, AK, USA
[3]Department of Geoscience, University of Oslo, Oslo, Norway
[4]Department of Geography, Rutgers, the State University of New Jersey, Piscataway, NJ, USA
[5]Department of Earth and Environmental Sciences, Ludwig Maximilian University, Munich, Germany
[6]Department of Geography and Regional Sciences, University of Graz, Heinrichstraße 36, 8010, Graz, Austria
[7]Department of Geosciences and Natural Resource Management (IGN), University of Copenhagen (UCPH), Øster Voldgade 10, DK-1350 Copenhagen, Denmark
[8]Geography, University of Liège, Liège, Belgium
[9]Institute for Marine and Atmospheric Research, Utrecht University, Utrecht, the Netherlands
[10]Department of Environmental Science, iClimate, Aarhus University, Roskilde, Denmark
[11]Danish Meteorological Institute, Copenhagen, Denmark

*Correspondence to*: Baptiste Vandecrux (bav@geus.dk)

## Abstract

The surface melting of the Greenland ice sheet has been increasing in intensity and extent over the last decades due to Arctic atmospheric warming. Surface melt depends on the energy balance which includes the atmospheric forcing but also the thermal budget of the snow, firn and ice near the ice sheet surface. The temperature of the ice sheet subsurface has been used as an indicator of the thermal state of the ice sheet's surface. We here present a compilation of more than 4612 measurements of ice, snow and firn temperature at 10 m below the surface ($T_{10m}$) across Greenland spanning from 1912 to 2022. The measurements are either instantaneous or monthly averages. We train an Artificial Neural Network model (ANN) on 4597 of these point observations, weighted by their relative representativity, and use it to reconstruct $T_{10m}$ over the entire Greenland ice sheet for the period 1950-2022. We use 10-year averages and mean annual values of air temperature and snowfall from the ERA5 reanalysis dataset as model input. The ANN indicates a Greenland-wide positive trend of $T_{10m}$ at 0.2 °C decade$^{-1}$ during the 1950-2022 period, with a cooling during 1950-1985 (-0.3 °C decade$^{-1}$) followed by a warming during 1985-2022 (+0.7 °C decade$^{-1}$). Regional climate models HIRHAM5, RACMO2.3p2 and MARv3.12 show mixed

results compared to the observational $T_{10m}$ dataset with mean differences ranging from -0.4 °C (HIRHAM) to 1.3 °C (MAR) and root mean squared differences ranging from 2.8 °C (HIRHAM) to 4.7 °C (MAR). The corresponding values for the ANN are -0.2 °C and 1.7 °C. The observation-based ANN also reveals an underestimation of the subsurface warming trends in climate models for the bare ice and dry snow areas. The subsurface warming brings the Greenland ice sheet surface closer to the melting point, reducing the amount of summer energy input required for melting. Our compilation documents the response of the ice sheet subsurface to atmospheric warming and will enable further improvements of models used for ice sheet mass loss assessment and reduce the uncertainty in projections.

1. **Introduction**

The Arctic is warming more than four times as fast as the global average (Chylek et al., 2022, Rantanen et al., 2022). Consequently, the Greenland ice sheet is exposed to an increase in air temperature (e.g. Hanna et al., 2021, Zhang et al., 2022) and increased anticyclonic, cloud free conditions in summer (Hofer et al., 2017; Ryan et al., 2022). In the low elevation bare ice area of the ice sheet, the warming atmosphere increases the heat transfer to the surface through turbulent heat fluxes (e.g., Wang et al. 2021), while a reduction in cloud cover increases the downward shortwave radiation, both resulting in melt increases since the late 1980s (Hofer et al., 2017, Trusel et al., 2018, Ryan et al., 2022). Enhanced melt in the bare ice area initiates multiple feedback processes, such as snowline retreat (Noël et al., 2019; Ryan et al., 2019) and algal growth (e.g. Stibal et al., 2017; Cook et al., 2020), which lead to further expansion and darkening of the bare ice area and enhanced shortwave radiation absorption. At higher elevations, increased surface melt also triggers a melt-albedo feedback through which liquid water within snow and grain coarsening decreases the snow albedo and increases the absorption of solar radiation (e.g. Nolin and Stroeve, 1997, Box et al., 2012). The increase in ice sheet surface energy influx leads to an increase in surface melt but also to an increase of subsurface temperatures through heat conduction and refreezing of meltwater (Humphrey et al., 2012, Polashenski et al., 2014, McGrath et al., 2013). The subsurface temperature is therefore a key indicator of how the Greenland ice sheet has been affected by recent climatic changes. Furthermore, ice sheet subsurface warming brings the near-surface snow and firn (multi-year, compressed snow) closer to the melting point and makes them less efficient at refreezing and retaining meltwater (Pfeffer et al., 1991; Vandecrux et al., 2020a). Subsurface warming could also trigger thermal regime shifts across the ice sheet (Marshall, 2021) and increase the ice viscosity (Phillips et al., 2010, 2013, Colgan et al., 2015) although with limited impact on dynamic mass loss (Poinar et al., 2017).

Over the last century, research teams have reported snow, ice and firn subsurface temperatures of the Greenland ice sheet. Of all depths measured, we here focus on measurements at, or close to, the 10 m depth. The temperature at this depth has been shown to be less affected by seasonal temperature variation and more representative of the long-term temperature and snowfall history at the surface (McGrath et al., 2013, Kjær et al., 2021). This makes it a convenient standard depth to compare temperatures from different periods and different sites. Here, we compile a dataset of 4612 observations of ice, snow and firn temperature at 10 m below the surface ($T_{10m}$) spanning from 1912 to 2022 from published and unpublished sources. We then use 4597 observations of $T_{10m}$ within the current ice sheet extent and the period 1950-2022 to train an Artificial Neural Network (ANN) model that can predict $T_{10m}$ over the entire ice sheet. For a given month and location, the ANN estimates $T_{10m}$ based on 14 parameters derived from the ERA5 reanalysis (Hersbach et al., 2020) that represent the long term and recent history of air temperature and snowfall. Using our observational dataset of subsurface temperature as well as our ANN, we evaluate three regional climate models (RCMs) widely used to estimate the surface mass balance of the Greenland ice sheet: RACMO2.3p2 coupled to an offline firn model IMAU-FDM v1.2G (hereafter RACMO, Noël et al., 2019, Brils et al., 2022), MARv3.12 (hereafter MAR, Fettweis et al., 2017, 2020) and HIRHAM5 (hereafter HIRHAM, Langen et al., 2017). We then evaluate the ANN and RCMs' $T_{10m}$ magnitudes and trends in the bare ice, percolation, and dry snow areas of the ice sheet. Lastly, we discuss the impact of this subsurface warming on the ice sheet mass balance processes.

## 2. Methods

### 2.1. Observed ice sheet subsurface temperature compilation and interpolation

A total of 4612 $T_{10m}$ observations were compiled from 48 sources (Figure 1, Table 1). Each dataset is described in the related reference in Table 1, except two yet undescribed datasets. The first unpublished dataset was collected by the late K. Steffen and his team and consists of two thermistor strings: one at Swiss Camp, central western Greenland, and another at Summit station, central Greenland, to complement the Greenland Climate Network (GC-Net) automated weather stations (AWS) at those sites (Steffen et al., 1996, 2001). The 11 m long string at Swiss Camp operated between 1992 and 2009 and was equipped with UUB thermistors at 0.5, 0.75 m depth and 1-11 m depth with 1 m spacing. The 10 to 15 m long string at Summit was equipped with Campbell Scientific T107 thermistors and was active during the periods 2000-2002 and 2007-2009. New sensors were added to the Summit string over the years. The sensors' depth and surface height evolution could be recovered from field

notes and this data is now presented for the first time. The second unpublished dataset comes from 14 new AWS installed in 2021 and 2022 by the Geological Survey of Denmark and Greenland (GEUS) as a continuation of the

GC-Net sites (Steffen et al., 1996, 2001, Vandecrux et al., 2023a). They are equipped with a GeoPrecision TNode thermistor string  with sensors installed at 0.5, 1, 1.5, 2, 2.5, 3, 4, 6, 8, 10 m depth. These data are hosted on the same dataset as the PROMICE AWS data (How et al., 2022).

We also post-processed two previously published datasets. The data from Humphrey et al. (2012) were corrected

for the changing depth of the sensor below the surface as snow accumulates or melts at the surface (Supplementary Text 1) - similar to the processing of the other time series. The FirnCover dataset (MacFerrin et al., 2022) appeared to have a warm bias due to the use of uncalibrated resistance temperature detectors instead of the conventional thermistor or thermocouple instruments. Using firn temperature observations reported by Samimi et al. (2021) and Heilig et al. (2018) at DYE-2 as a reference, we built an ad-hoc correction function that

was then applied at all sites within the FirnCover dataset. The correction procedure is described in Supplementary Text 2 and reduces the FirnCover temperatures by 1.1 °C on average.

For the temperatures continuously recorded by thermistor or thermocouple strings, the depth of each temperature sensor below the surface were calculated using installation depths and recorded surface height. Wherever

necessary, we interpolated the available temperature profiles linearly to 10 m depth and allowed linear extrapolation if at least two measurements were available within 2 m of the 10 m depth. The resulting $T_{10m}$ values were then aggregated as monthly means if they originated from continuous measurements or left as instantaneous values otherwise.

The measurements conducted by different scientific teams at the same location allow for an assessment of uncertainty and reproducibility of "local" vertically interpolated $T_{10m}$ observations. From 10 sites where simultaneous measurements are available, the median root mean square difference (RMSD) is 0.5 °C (Supplementary Table 1). Among these 4612 $T_{10m}$ observations, 15 measurements are either outside of the current ice sheet extent as defined by the GIMP ice sheet delineation (Howat et al., 2014) or outside of the 1950-2022

period we consider for our $T_{10m}$ reconstruction. There are therefore 4597 $T_{10m}$ observations in our compilation that can be used for the reconstruction of $T_{10m}$ on the ice sheet between 1950 and 2022.

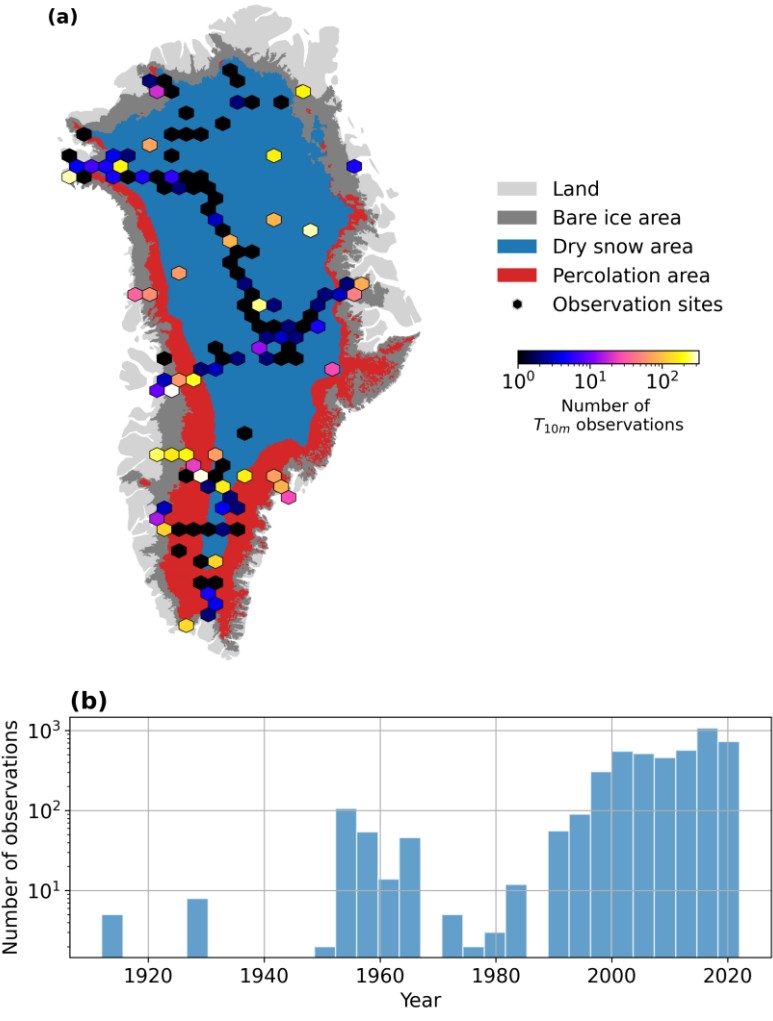

**Figure 1. Spatial (a) and temporal (b) distribution of the T$_{10m}$ observations in Greenland. Greenland surface classification according to Vandecrux et al. (2019) based on firn density profiles and remote sensing observations.**

**Table 1. Overview of T$_{10m}$ datasets used in this study.**

| Reference | Start year | End year | Number of measurements |
|---|---|---|---|
| Koch (1913) | 1912 | 1913 | 5 |
| Wegener (1930); Abermann et al. (2023) | 1930 | 1930 | 8 |
| Heuberger (1954) | 1950 | 1950 | 2 |
| Benson (1962) | 1954 | 1955 | 59 |
| Schytt (1955) | 1954 | 1954 | 31 |

| | | | |
|---|---|---|---|
| Nobles (1960) | 1954 | 1954 | 7 |
| Heuberger (1954) | 1954 | 1954 | 1 |
| Meier et al. (1957) | 1955 | 1955 | 4 |
| Griffiths (1960) | 1955 | 1956 | 38 |
| de Quervain (1969) | 1957 | 1964 | 8 |
| Ambach (1979) | 1959 | 1959 | 2 |
| Langway (1961) | 1959 | 1959 | 14 |
| U.S. Army Transportation Board (1960) | 1960 | 1960 | 4 |
| Davis (1954) | 1960 | 1960 | 7 |
| Davis (1967) | 1962 | 1962 | 1 |
| Mock (1965) | 1964 | 1964 | 12 |
| Mock and Ragle (1963) | 1964 | 1964 | 31 |
| Weertman et al. (1968) | 1966 | 1966 | 1 |
| Colbeck and Gow (1979) | 1973 | 1973 | 3 |
| Clausen et al. (1988) | 1974 | 1985 | 11 |
| Clausen and Hammer (1988) | 1977 | 1977 | 1 |
| Stauffer and Oeschger (1979) | 1978 | 1978 | 3 |
| Clement (1984) | 1983 | 1983 | 4 |
| Thomsen et al. (1991) | 1990 | 1991 | 8 |
| Ohmura et al. (1992) | 1990 | 1990 | 3 |
| GC-Net unpublished | 1991 | 2010 | 170* |
| Braithwaite (1993) | 1991 | 1992 | 12 |
| Laternser (1994) | 1992 | 1992 | 16 |
| Schwager (2000) | 1994 | 1994 | 1 |
| Historical GC-Net: Steffen et al. (1996, 2001, 2023); Vandecrux et al. (2023a) | 1995 | 2022 | 1662* |
| Giese and Hawley (2015) | 2004 | 2008 | 47* |
| Humphrey et al. (2012) | 2007 | 2009 | 57* |
| PROMICE: Fausto et al. (2021); How et al. (2022) | 2008 | 2022 | 1315* |
| Smeets et al. (2018) | 2009 | 2016 | 160* |
| Harrington et al. (2015) | 2010 | 2012 | 5 |
| Hills et al. (2018) | 2011 | 2017 | 109* |
| Charalampidis et al. (2016) ; Charalampidis et al. (2022) | 2012 | 2013 | 29* |
| Yamaguchi et al. (2014) | 2012 | 2012 | 1 |
| Miller et al. (2020) | 2013 | 2017 | 68* |

| | | | |
|---|---|---|---|
| Polashenski et al. (2014) | 2013 | 2013 | 2 |
| Matoba et al. (2015) | 2014 | 2014 | 1 |
| MacFerrin et al. (2021, 2022) | 2015 | 2019 | 311* |
| Kjær et al. (202121) | 2015 | 2015 | 2 |
| Heilig et al. (2018) | 2016 | 2021 | 58* |
| Vandecrux et al. (2021); Colgan and Vandecrux (2021) | 2017 | 2022 | 119* |
| Covi et al. (2022, 2023) | 2017 | 2019 | **77** |
| Law et al. (2021) | 2019 | 2019 | 1 |
| GC-Net continuation: Fausto et al. (2021); How et al. (2022) | 2021 | 2022 | 121* |
| | | **Total:** | **4612** |

\* monthly mean values derived from the original measurements

### 2.2. **The artificial neural network**

Point observations of $T_{10m}$ only give a partial description of the subsurface temperature: they are discontinuous in space and time. To describe the evolution of $T_{10m}$ over the entire ice sheet and over the last decades, one can train a machine learning model that links $T_{10m}$ to an input dataset which is itself continuous in space and time and assumed to drive changes in $T_{10m}$. Once the relationship between input and $T_{10m}$ is learned by the algorithm, the algorithm can be driven by the entire input dataset to reconstruct the $T_{10m}$ even at places where no observations are available.

Among machine learning algorithms, ANNs have proven their ability to learn non-linear relationships between a target variable and a set of input variables in numerous glaciological and meteorological applications (e.g. Steiner et al., 2005, Braakmann-Folgmann and Donlon, 2019, Xu et al., 2021). Given that our $T_{10m}$ compilation, which will be used to train the algorithm, does not encompass all possible ice sheet conditions, we favor ANNs over tree-based algorithms that can be limited when used beyond their training dataset (e.g., Xiong et al., 2020, Liu et al., 2022). Finally, during our search for the most straightforward ANN structure capable of modeling our dataset, we ultimately chose a multi-layer perceptron (Rumelhart et al. 1986). We want to highlight that the choice of model structure, input parameters and training strategy does not have a single optimal configuration. Some of our choices are even decreasing the apparent performance of the model in order to avoid overfitting and to increase the model's capacity to extrapolate outside of its training set. The following sections describe selection of inputs, our enhancement of the dataset's representativity and eventually the ANN structure, training and uncertainty assessment.

### 2.2.1. **The input parameters**

Our target variable, $T_{10m}$, is predominantly controlled by 1) the surface temperature through molecular heat conduction, 2) the subsurface refreezing of meltwater through latent heat release, and 3) snowfall rates which determine the vertical advection velocity in the firn column. The near-surface air temperature can act as a proxy for both surface temperature and surface melt in the absence of reliable estimates, because they all interact within the surface energy budget. The surface temperature itself depends on the near-surface air temperature through

turbulent heat fluxes and the surface energy budget. This relationship between air temperature, snowfall and $T_{10m}$ is notably non-linear. In regions where surface melt is common, meltwater refreezing at depth will lead to $T_{10m}$ several degrees higher than the average air temperature (e.g. Humphrey et al, 2012). On the other hand, during periods of minimal or no melting (wintertime or nighttime in the summer), the radiative imbalance at the surface and the presence of a near-surface atmospheric temperature inversion can cause the surface temperatures, and

through conduction the T10m, to be several degrees lower than the near-surface air temperature (e.g. Miller et al., 2017, Steffen and Box, 2001). Additionally, snowfall affects the subsurface temperature in several ways. In the ablation area, the seasonal snowpack insulates the underlying ice. In the accumulation area, snow accumulated at the surface is, after some time, advected to greater depth, where it can act as either a heat source or sink depending on its temperature at time of deposition.

We here use the air temperature and snowfall monthly grids from the ERA5 reanalysis (Hersbach et al., 2020) to derive our 14 input parameters. We use ERA5 Land at spatial resolution 0.1x0.1 ° for 1950-2022 (Muñoz Sabater, 2019) and the original ERA5 (Hersbach et al., 2023) at 0.25x0.25 ° resolution resampled linearly to 0.1x0.1 ° for 1940-1950. Delhasse et al. (2020) showed that daily ERA5 near-surface air temperatures compare well with measurements from ice-sheet weather stations (mean bias of 0.01 °C, root mean square error of 3.05

°C). Loeb et al. (2022) found that ERA5's precipitation had the best performance out of three evaluated reanalysis datasets against weather station observations in the Canadian Arctic and in Greenland. Using airborne radar measurements of snow accumulation, Ryan et al. (2020) found that ERA5's annual snowfall in Greenland was comparable to estimates from state-of-the-art RCMs and outperformed satellite estimations.

The 10 year average temperature ($\overline{T_{2m,10\,y}}$) and snowfall ($\overline{SF_{10\,y}}$) were calculated for each cell and each month

to represent the long term conditions at a given time and place. Additionally, for each grid cell and monthly time step we calculate the amplitude of the 2 m air temperature in the previous year ($T_{2m,\,amp}$) as well as the average air temperature and snowfall of the five previous years. This reflects the capacity of the subsurface to respond, not only to long term changes, but also to recent changes in air temperatures and snowfall (e.g. Polashenski et al.,

2014). Lastly, to assist the ANN in capturing the annual periodicity, we give as input the cosine of the month
(assigning 1 in January and -1 in July). For a given time and location, the ANN therefore takes 14 inputs:
$\overline{T_{2m,10\,y}}$, $\overline{SF_{10\,y}}$ and $T_{2m,\,amp}$, the five previous years of annual snowfall, the five previous years of air temperature
and the month's cosine.

### 2.2.2. **Weighting of the observations prior to ANN training**

Many machine learning algorithms, including ANNs, assume that the training data are representative of the target
area (where the model is applied for predictions), i.e., that the data are drawn from the same distribution. This
assumption is violated in practice when applying the model to new spatial domains that may contain local
conditions not present in the training data. Thus, the representativity of the training dataset compared to the target
area is critical for the robustness of any machine learning model, i.e. how well the model generalizes to new and
unseen data (Meyer and Pebesma, 2021; Bjerre et al., 2022). The representativity of the 4597 observation sites
(training data) compared to the entire Greenland ice sheet where the ANN is applied (target area) was quantified
using histogram analysis (Figure 2). For the three input parameters that define the climate at a given location
($\overline{T_{2m,10\,y}}$, $\overline{SF_{10\,y}}$ and $T_{2m,\,amp}$), here referred to as $p_{i\,=\,1,\,2,\,3}$ , we plot the probability histogram of the parameter $p_i$
as it appears in ERA5 at our observation locations: this is the 'observation' histogram $H_o(p_i)$. We then plot, for
that input parameter $p_i$, the probability histogram of all the ice sheet pixels, and all time steps within the ERA5
dataset: this is the 'target' histogram $H_t(p_i)$. The 'observation' histograms $H_o(p_i)$ represent the distribution the
ANN will learn from while the 'target' histograms $H_t(p_i)$ represent the values over which the ANN will
eventually be applied (Figure 2). In an ideal scenario where the observational dataset is representative of the
parameter space where the ANN will be applied, $H_o(p_i)$ and $H_t(p_i)$ should show similar distributions.

In practice, the available observations are not representative for the entire ice sheet steming from, e.g.,
monitoring sites producing data continuously or western Greenland being more accessible than eastern
Greenland. To make the training dataset more representative of the parameter space in which the ANN will be
used, we define for each observation a weight $w_{obs}$ as follows. For each observation and for a given input
parameter $p_i$, $w_{obs}(p_i)$ is equal to the ratio of $H_t(p_i)$ and $H_o(p_i)$ for the bin where the observation is located.
Consequently, if in a given bin, the observation histogram is lower than the target histogram, meaning that this
bin is underrepresented in the observational dataset compared to the target space, then the weight $w_{obs}(p_i)$ will be
greater than one. InverselyI, the weight $w_{obs}(p_i)$ will be less than one if the observation histogram is greater than
target histogram. Eventually, for each observation, we calculate the overall weight $w_{obs}$ as the mean of $w_{obs}(p_1)$,

$w_{obs}(p_2)$ and $w_{obs}(p_3)$. This overall weight $w_{obs}$ for each observation is used to calculate the loss function (in our

case the mean squared error) minimized during the training of the ANN. As a consequence, observations that are located in underrepresented regions of the parameter space will have overall weights $w_{obs} > 1$ and will be given more importance in the training of the ANN. Inversely, observations located in underrepresented parts of the parameter space will have overall weights $w_{obs} < 1$ and will count less in the training of the ANN.

As an illustration, let us consider a $T_{10m}$ observation from a site and time that has $\overline{T_{2m,10\,y}}$ = -28°C. Figure 2a indicates that only ~10% of our observation sites have such an average temperature, compared to ~23% of the ice sheet pixels in ERA5, i.e., this sample comes from an under-represented temperature range. Following our procedure, we allocate to this observation $w_{obs}(p_1) = 0.23/0.1 = 2.3$ to increase its final weight $w_{obs}$, which also considers the observation's representativity with regard to $\overline{SF_{10\,y}}$ and $T_{2m,\,amp}$. Inversely, 25 % of our observation

have $\overline{T_{2m,10\,y}}$ = -18°C while only 10% of the ice sheet (according to ERA5) has such average temperature (Figure 2a). Consequently, an observation having such $\overline{T_{2m,10\,y}}$ will receive a $w_{obs}(p_1) = 0.1/0.25 = 0.4$ and will weigh less in the training of our ANN.

To verify that our weighting procedure increases the similarity between $H_o(p_i)$ and $H_t(p_i)$, we evaluate the

distance between two histograms $H_1$ and $H_2$ calculated on the same $n$ bins with the Canberra distance (Lance and Williams, 1966, Emran and Ye, 2001):

$$d_{Canberra}(H_1, H_2) = \sum_{k=1}^{n} \left(\frac{|H_1(k) - H_2(k)|}{H_2(k)}\right)$$

where $H_{i=1,2}(k)$ is the value of histogram $H_{i=1,2}$ at bin $k$. The smaller the Canberra distance $d_{Canberra}(H_o(p_i), H_t(p_i))$, the more $H_o(p_i)$ and $H_t(p_i)$ are similar. The Canberra distance between observational

and target histogram for $\overline{T_{2m,10\,y}}$, $\overline{SF_{10\,y}}$ and $T_{2m,\,amp}$ decreased from 22.6, 12.2 and 14.3 to 11.1, 7.5 and 5.3 when weighing the observations based on their representativity (Figure 2). Another confirmation that the weights increase the similarity between the observation and target histograms is the smaller difference between the observation and target distributions' median values once the weights are applied: from 4.9 °C, 6.0 mm w.e., 2.4 °C with equal weights (Figure 2a-c) to 2.1 °C, 1.4 mm w.e. and 0.4 °C with weights (Figure 2d-f), for $\overline{T_{2m,10\,y}}$,

$\overline{SF_{10\,y}}$ and $T_{2m,\,amp}$ respectively.

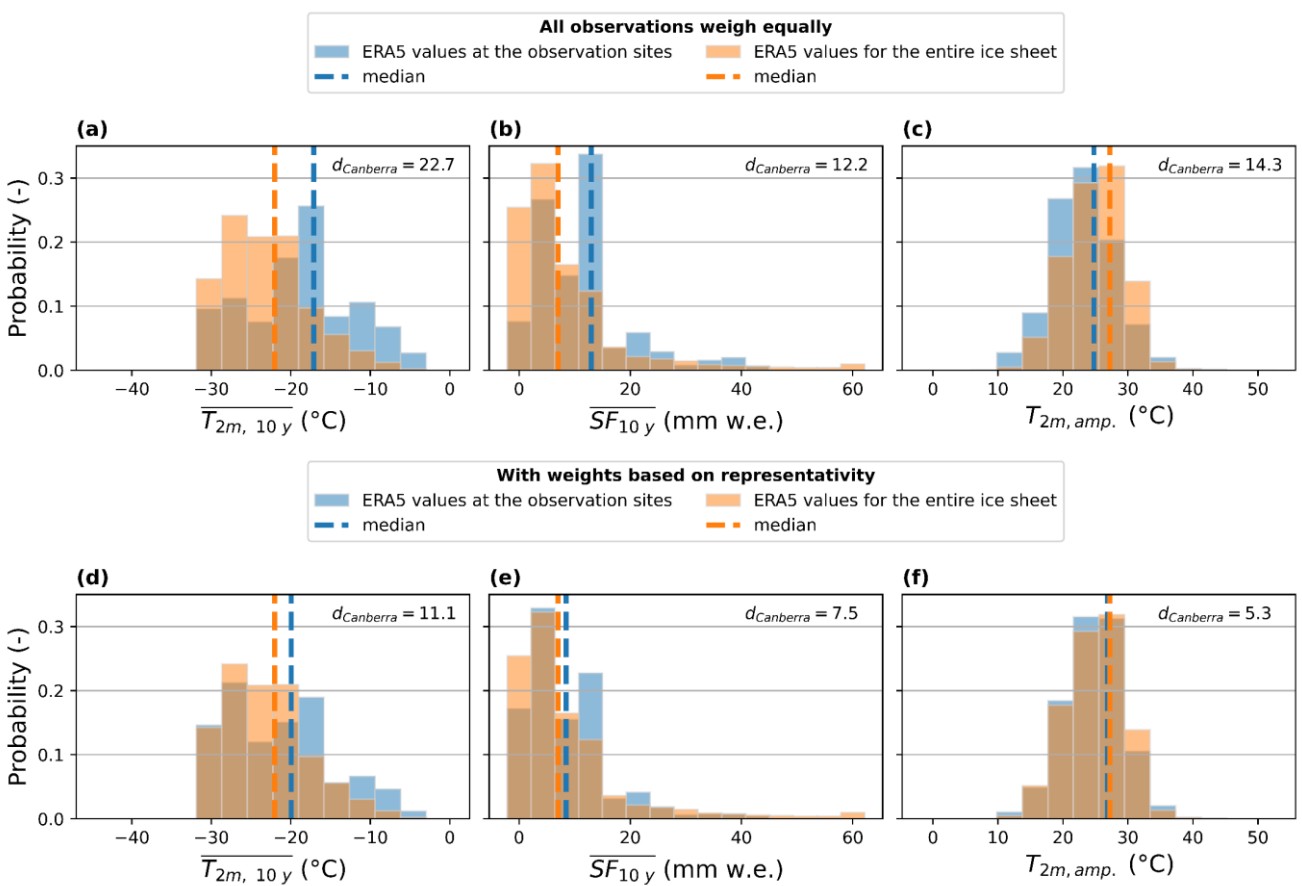

**Figure 2: Histograms of the input parameters: 10 year average 2 m air temperature, 10 year average snowfall, annual amplitude of monthly 2 m air temperature. The blue histograms are parameter values as they appear at the observation sites, meaning the training data for the ANN, either with all observations weighing the same (a,b,c) or with weights assigned to each observation based on its representativity (d, e, f). The orange histograms are parameter values as they appear in the ice sheet pixels of ERA5, meaning the target data for the ANN. For each pair of target and observation histograms, we calculate the Canberra distance ($d_{Canberra}$) as a measure of similarity.**

### 2.2.3. ANN structure and training

Multilayer perceptron ANNs are typically composed of an input layer, with as many nodes as input variables, multiple hidden layers containing several nodes, and an output layer. Each node in the hidden layers: i) makes the weighted sum of the outputs of all nodes from the preceding layer and adds a node-specific bias, ii) applies a simple, layer-specific activation function to the result, and iii) passes the output of the activation function to all the nodes of the next layer, and so forth. During the training of the model, all the weights and biases from all the nodes are being optimized to minimize a loss function. This is done iteratively by: i) passing part of the training

set through the ANN, ii) evaluating the difference between the ANN output and the expected result using the loss function, and iii) updating the weights and biases to reduce the error in the next iteration (a.k.a. backpropagation). This general ANN structure can be adapted in many ways to the dataset and problem it is applied to. Here, we adjust four of the most important hyperparameters of the ANN: the batch size, i.e. which fraction of the sample is given to the ANN for every training cycle; the number of epochs (or training cycles); the number of layers and

the numbers of nodes within those layers. We use the Adam optimizer (Kingma and Ba, 2014), rectified linear unit activation function (0 if the input is below 0, $f(x)=x$ if the input x is above 0) and mean squared error as loss function. Those three settings have been used widely in regression problems (Braakmann-Folgmann and Donlon, 2019; Liu et al., 2022, Lorentzen et al., 2022).

We set the hyperparameters of our ANN in three steps. First, we define a validation set made of 633 observations (14% of the training dataset) from four sites representing different areas of the ice sheet: NASA-E for the dry snow area, NASA-SE for the percolation area and Swiss Camp and KAN_M for the bare ice area, and use these data as a validation set. Secondly, we train an ensemble of ANNs with two layers of 32 nodes each with batch sizes varying from 100 to 5000 (18 irregular steps) and between 10 and 1000 epochs (8 irregular steps). Each of

the 144 settings are being run 10 times to account for the stochastic processes within model training, resulting in a total of 1440 ANNs. We assess the average learning curve for each setting: the mean difference (MD) and root mean squared difference (RMSD) of the trained ANN on the training and validation data as a function of epoch numbers (Supplementary Figure 1). We conclude that: i) small batch sizes (<1000) lead to unstable learning curves (Supplementary Figure 1a-d) and ii) large batch sizes (e.g. 5000) cause slightly slower convergence and

similar results as batch sizes of 3000 and 4000 (Supplementary Figure 1i-n). From our analysis and as a compromise between stability, rapidity of convergence and potential overfitting, we use a batch size of 4000 over 150 epochs for all ANN trained henceforth. In the third and last step of our hyperparameter tuning, we use the optimal batch size and number of epochs to train 180 ANNs with either 1, 2 and 3 layers of 8, 16, 32, 64, 128 and 256 nodes each (all layers with same number of nodes, each setting repeated 10 times). We see clear

improvements (lower RMSD) when moving from a single layer to two layers, and from 8 nodes to 64 nodes (Supplementary Figure 2). The improvement moving from 2 to 3 layers and from 64 to 128 or 256 nodes are marginal and within the stochastic uncertainty (overlapping standard deviations in Supplementary Figure 2c-f). To keep the model design as simple as possible, we henceforth use two layers of 64 nodes each.

Additionally, a Gaussian noise layer that adds random noise to the observations is added after the input layer to further prevent overfitting (e.g. An, 1996). Note that both the addition of Gaussian noise and the assignment of weights to the observations will tend to decrease the apparent performance of the ANN (e.g. MD or RMSD from the non-weighted observational dataset) but will produce a more robust output and prevent overfitting. Considering the limited number of observations relative to the target area, the entire Greenland ice sheet, we train

our "best model" using all the available observations weighted according to their representativity. Consequently, there is no  hold out, or unseen data for model validation. Alternatively, we use a spatial cross-validation approach to measure the performance and uncertainty of the ANN.

### 2.2.4. **Uncertainty estimation of the ANN with spatial cross-validation**

Spatial cross-validation is considered the best-practice approach for evaluating the uncertainty of ANN when
dealing with spatial data (e.g. Brenning et al., 2012). For this purpose, we separated the Greenland ice sheet into 10 regions (Figure 3c) after Zwally et al. (2012). Each of the 10 regions contain between 95 and 1298 observations, corresponding to 2% and 28% of all observations. For 10 iterations, we hold out the observations located in a different region and train an ANN on the remaining observations. We save these 10 models and for any new set of input parameters, we use the standard deviation of the 10 models' predictions as a measure of the
uncertainty. This uncertainty is never allowed to be below 0.5 °C, which is the measurement uncertainty derived in Section 2.1. The monthly grids of ANN uncertainty are provided along with our best estimation of $T_{10m}$, which is produced by an ANN trained on all available observations.

For a fair evaluation of our ANN against our observational dataset, we first compare our best ANN model,
trained on all $T_{10m}$ observations to these same $T_{10m}$ observations. This evaluation does not show how the model would perform on new, unseen data or regions, and consequently leads to an overestimation of the ANN performance. We then compare each $T_{10m}$ observation to the corresponding $T_{10m}$ predicted by the one cross-validation model that did not use this observation for training. This second evaluation illustrates how the cross-validation ANNs perform on data that was not included in the training set. It contrasts with the first assessment,
because it evaluates models that were trained only on part of the observation dataset, and it is therefore a conservative estimate of the performance of the best model trained on all $T_{10m}$ observations.

## 2.3. Regional climate models

We evaluate 10 m subsurface temperatures from three regional climate models: MARv3.12 (Fettweis et al., 2017, 2020), RACMO2.3p2 (Noël et al., 2019) with the updated IMAU-FDMv1.2G (Brils et al., 2022) and HIRHAM5
(Langen et al., 2017). We calculate the MD and RMSD between the observed and simulated 10 m subsurface temperatures. For this study, the output from MAR, RACMO and HIRHAM are available over the periods 1950-2020, 1958-2020 and 1980-2016, respectively. We compare each model to the measurements within the common 1980-2016 period for which all three model outputs are available, as well as against all observations.

All three models use a multilayer snow, firn and ice model to calculate subsurface temperature. In addition to
differences in surface forcing in the three models (e.g. in snowfall, rainfall, melt and energy fluxes), the models also differ in the way they calculate the subsurface characteristics that impact the subsurface temperature. Both MAR and HIRHAM estimate firn densification using the overburden pressure: respectively from Brun et al. (1989) and Vionnet et al. (2012); while RACMO uses a compaction law that was derived for steady-state firn (Arthern et al., 2010) and empirically fitted to observations (Ligtenberg et al., 2011; Brils et al., 2022).
RACMO's offline run with IMAU-FDMv1.2G uses the thermal conductivity parameterization from Calonne et al. (2019) while HIRHAM and MAR use the parameterization by Yen (1981). The three models treat the release of latent heat during the refreezing of meltwater in a similar manner, but the meltwater infiltration is calculated differently. Both MAR and RACMO use a bucket scheme: meltwater infiltrates downward unless the water is refrozen or retained through capillary forces and ice layers are considered permeable at the model scale
(Ligtenberg et al., 2018). In HIRHAM, the use of a parameterization of Darcy flow (Hirashima et al., 2010) and accounting for the decrease of the layer permeability due to ice content (Colbeck, 1975) lead to shallower infiltration than in RACMO (Vandecrux et al., 2020b). Another model detail that impacts the calculated subsurface temperature is the boundary condition at the bottom of the model domain. HIRHAM uses a temperature scheme that requires a fixed temperature at the lowermost firn layer which is set, for each pixel, to
the long term air temperature average (Langen et al., 2017). Both MAR and RACMO use the Neumann boundary condition at the bottom layer of the firn model, which implies no heat flux through the lower boundary of the model. However, in the ablation area, new material needs to be provided to the bottom layer of the model as surface ablation melts ice away. In MAR, as soon as the model column height is lower than 29 m, a 1 m thick layer composed of ice is added at the bottom of the model column. MAR then uses a simple assumption that the
underlying ice would always be cooler than the ablating, near-surface ice. Consequently, the temperature of the 1 m layer added at the bottom of the model was fixed to be 1% lower (when calculated in Kelvin) than the

temperature of the lowermost layer left in the model. The differences between RCM-simulated subsurface temperatures are partly due to these different modeling approaches for the subsurface processes. This can be illustrated when different subsurface models are forced with similar surface data (Lundin et al., 2017; Vandecrux

et al., 2020b). Another source of discrepancy is the difference in surface climate that is simulated in each of these three models. More information about the accuracy of the simulated surface climate within each RCM can be found in Fettweis et al. (2020), Langen et al. (2017) and Noël et al. (2019).

3.  **Results**

3.1.  **Performance of the ANN**

When comparing the best ANN model to the $T_{10m}$ observations it was trained on, we find a MD of 0.0°C and a RMSD of 1.6 °C (Figure 3a). However, when evaluating the cross-validation models against their respective unseen data, we find a similar MD (0.1ºC) and a RMSD of 2.5 °C (Figure 3b). While the first evaluation is overoptimistic, the second does not directly evaluate our best ANN model, which is trained on all available data. These estimates nevertheless provide bounds to the true performance of our ANN.

Averaging over the entire period 1950-2022, the ANN uncertainty is lowest across the dry snow area (Figure 3c), illustrated by the NASA-E site (Figure 3d). The uncertainty increases towards the ice sheet margin in west, north and northeast Greenland (Figure 3c) as exemplified by the sites DYE-2 in the percolation area in western Greenland (Figure 3e), KAN_L and KPC_U (Figure 3f-g), two ablation area sites in western and northeastern Greenland, respectively. The ANN uncertainty peaks in southeast Greenland (Figure 3c) where relatively high air

temperatures and snow accumulation produce temperate firn conditions and firn aquifers (Forster et al., 2013; Kuipers Munneke et al., 2014). When the measurements in this region are removed for cross-validation, there are no firn aquifer observations left in the training set for the ANN to learn what the $T_{10m}$ structure in this ice sheet region is. This is illustrated in Figure 3h by a cross-validation model predicting lower $T_{10m}$ than observed at the site FA_13 resulting  in a larger standard deviation between the cross-validation models for FA_13. Our

uncertainty estimation is conservative because the final ANN model is eventually trained on all observations. The regions of high uncertainty highlight where observations are the most needed to map the ice sheet subsurface temperature.

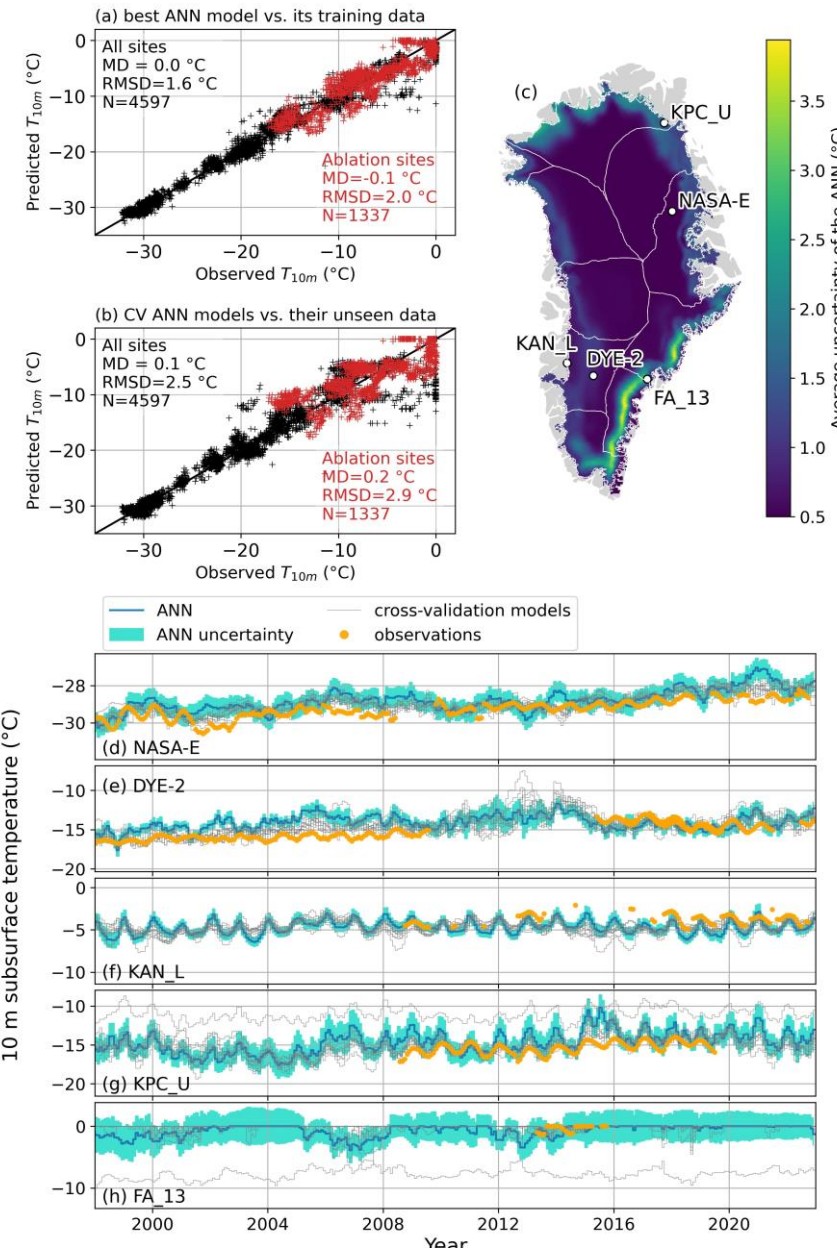


**Figure 3. (a) Evaluation of the T$_{10m}$ simulated by the best ANN model against the observations used for training. (b) Evaluation of the T$_{10m}$ simulated by the 10 cross-validation (CV) ANN models against their unseen data (i.e. not used for training). The statistics presented are mean difference (MD), root mean square difference (RMSD) and number of samples for which the comparison was possible (N). (c) 1950-2022 average of the ANN uncertainty as calculated from the standard deviation of 10 cross-validation ANN models trained on different spatial subsets of the observation dataset. (d-h) Examples of ANN T$_{10m}$ prediction, its uncertainty and the prediction of the 10 cross-validation models at a dry snow site (NASA-E), one percolation site (DYE-2), two ablation sites (KAN_L, KPC_U), and a firn aquifer site (FA_13).**

To evaluate the capacity of the ANN to capture the recent evolution of $T_{10m}$, we select 10 sites where more than 60 monthly values are available between 1998 and 2022 and compare the trends calculated from the ANN and the observations over the periods 1998-2010 and 1998-2022 (Table 2). These periods were chosen because of a general lack of measurements between 2011 and 2020. Trends calculated from the ANN only consider the months where observations are available. We note that due to the missing months, these trends are not reliable for general inference on the true $T_{10m}$ evolution: depending on which months are missing it might overestimate or underestimate the true $T_{10m}$ trend for these periods. The median $T_{10m}$ trends for 1998-2010 are 0.9 and 0.8 °C decade$^{-1}$ for the ANN and for the observations respectively (Table 2). For the period 1998-2022, the median $T_{10m}$ trends for 1998-2010 are 0.4 and 0.6 °C decade$^{-1}$ for the ANN and for the observations respectively (Table 2). The ANN therefore slightly overestimates the $T_{10m}$ trend during 1998-2010 and underestimates it during 1998-2022. We conclude that the ANN reproduces the magnitude of the $T_{10m}$ increase seen in observations although this aptitude varies with the location and the time period considered. From this assessment and because the ANN does not suffer temporal nor spatial gaps, the ANN appears as a suitable tool to study the trends in $T_{10m}$ over the entire Greenland ice sheet.

**Table 2: Trends in 10 m subsurface temperature ($T_{10m}$) calculated from the ANN and observations (obs.) at 10 sites for the periods 1998-2010 and 1998-2022. ANN trends are calculated only from the months where observations are also available. The difference between the two calculated trends as well as the number of monthly values used for the calculation (N) are also given for each site.**

| | Trends in T10m (°C decade-1) | | | | | | | |
| | 1998-2010 | | | | 1998-2022 | | | |
| Site | ANN | obs. | ANN - obs. | N | ANN | obs. | ANN - obs. | N |
|---|---|---|---|---|---|---|---|---|
| NASA-SE | 1.0 | 0.7 | 0.3 | 115 | 0.4 | 0.5 | -0.1 | 171 |
| NASA-E | 0.5 | 0.5 | 0.1 | 140 | 0.6 | 0.5 | 0.0 | 270 |
| Summit | 0.4 | 1.0 | -0.6 | 133 | 0.3 | 0.6 | -0.3 | 172 |
| Tunu-N | 0.7 | 0.3 | 0.4 | 140 | 0.6 | 0.5 | 0.0 | 150 |
| South Dome | 1.4 | 0.8 | 0.5 | 97 | 0.2 | 0.5 | -0.2 | 116 |
| Saddle | 1.4 | 0.7 | 0.7 | 125 | 0.2 | 0.6 | -0.4 | 156 |
| Humboldt | 0.5 | 1.0 | -0.4 | 66 | 0.4 | 0.3 | 0.1 | 71 |
| Crawford Point 1 | 1.3 | 3.0 | -1.7 | 63 | 0.4 | 0.7 | -0.3 | 120 |
| DYE-2 | 1.2 | 0.8 | 0.4 | 139 | 0.3 | 1.1 | -0.7 | 220 |
| Swiss Camp | 0.7 | 0.8 | 0.0 | 83 | 0.3 | 1.8 | -1.5 | 172 |

## 3.2. RCM evaluation and comparison with the ANN

We evaluate the RCMs against the observed $T_{10m}$ in the period 1980-2016 for which all three RCM's outputs are available (Figure 4). HIRHAM shows the best performance (MD = -0.4 °C, RMSD = 2.8 °C), followed by RACMO (MD = -1.3 °C, RMSD = 3.1 °C) and MAR (MD = +1.2 °C, RMSD = 4.7 °C). For the observation sites located in the ablation area, RACMO, HIRHAM and MAR have a cold bias with MD of -3.6, -0.9 and -3.4 °C respectively (Figure 4). MAR captures neither the geographical nor the seasonal variability of $T_{10m}$ in the ablation area (RMSD = 5.4 °C). The ANN, although of a different nature, gives better statistics at these ablation sites with a MD of 0.2 °C and a RMSD of 2.9 °C, even when calculated from our cross validation models' unseen data (Figure 3b).

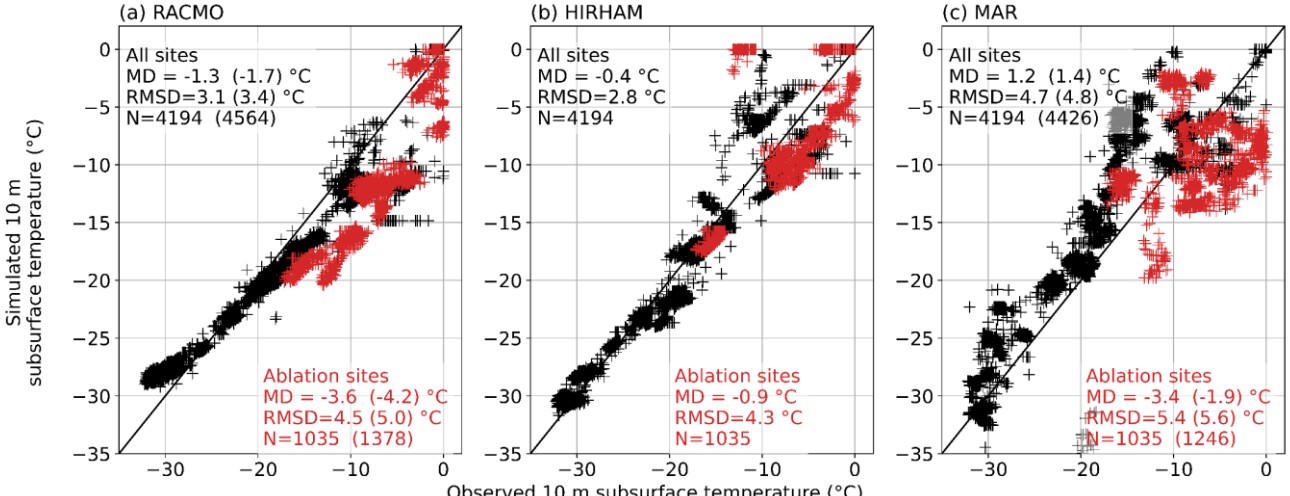

**Figure 4. Evaluation of the monthly 10 m subsurface temperatures simulated by RACMO (a), HIRHAM (b) and MAR (c) against observations. The statistics presented are mean deviation (MD), root mean square difference (RMSD) and number of samples for which the comparison was possible (N) for the period when all three models are available (1980-2016). For RACMO and MAR, the statistics for all available measurements are given in the parenthesis. For sites where annual surface ablation is larger than snow accumulation, i.e., net ablation sites with a bare ice cover in summer, symbols and statistics are shown in red.**

We further evaluate the ANN and RCMs at eight sites (Figure 5) that are representative of the dry snow (Summit, NASA-E), percolation (DYE-2, KAN_U), bare ice (Swiss Camp, KPC_U, SCO_U) and firn aquifer regions (FA_13). The ANN performs well at most of these sites: the average MD for these eight sites is less than 0.2 °C and the average RMSD is 1.2 °C. RACMO overestimates $T_{10m}$ at lower temperature sites in the dry snow area (Figure 5a-b) and underestimate $T_{10m}$ at the accumulation sites with relatively high melt (Figure 5c-d) and at ablation sites (Figure 5e-g). HIRHAM compares better than RACMO to the measurements at accumulation sites (Figure 5a-b) and can either over- or underestimate $T_{10m}$ at percolation sites (Figure 5c-d) and at ablation sites

(Figure 5e-g). MAR simulates $T_{10m}$ that are unrealistic both in magnitude and in variations (Figure 5). The causes of this low performance will be discussed in Section 4. At a firn aquifer site (Figure 5h) the ANN and the three RCMs successfully estimate relatively high $T_{10m}$ during the period 2013-2015 for which observations are available. Yet, the models diverge significantly when estimating the past history of the site: HIRHAM and MAR indicate $T_{10m}$ close to 0 °C from the models' respective initiations in 1980 and 1950, while RACMO and the ANN indicate that $T_{10m}$ below -2 °C may have been common at FA_13 before 2000 (Figure 5h).

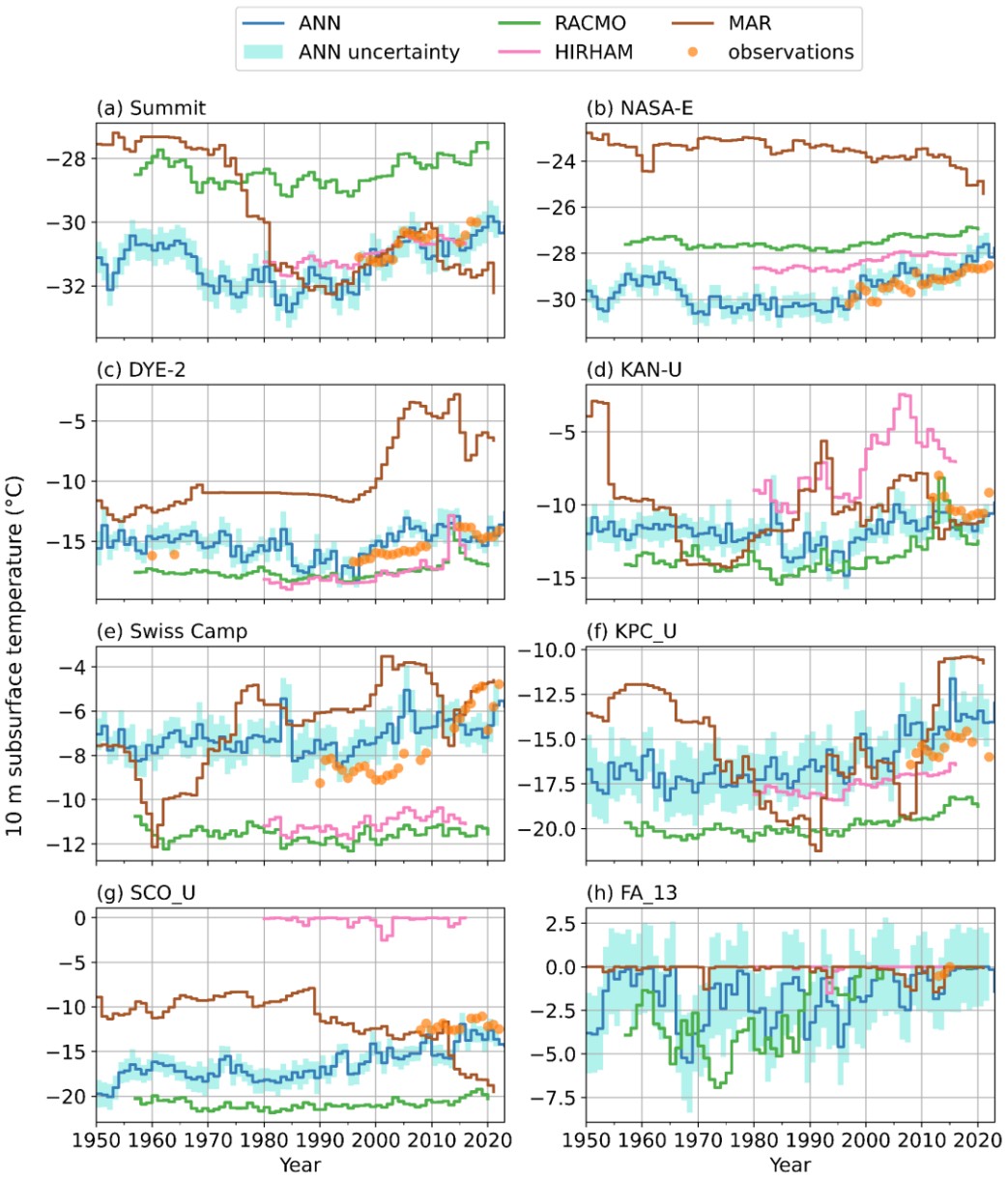

**Figure 5. Observed and simulated 10 m subsurface temperatures at selected sites. Note the different y-axese.**

### 3.3. $T_{10m}$ trends in the ANN and RCMs

According to the ANN, the Greenland ice sheet average $T_{10m}$ has been increasing significantly at a rate of +0.2 °C decade$^{-1}$ (P<0.01) over the 1950-2022 period (Table 3, Figure 6a), from an ice sheet-wide average value of -21.1 °C in 1950 to -19.2 °C in 2022. This increase was not constant over the 1950-2022 period. When fitting multiple piecewise linear functions to the Greenland ice sheet average $T_{10m}$, with breakpoint between 1951 and 2021, we identify 1985 as the breakpoint year that explains most of the variance in the ice-sheet-wide average $T_{10m}$ time series. This piecewise linear function consists of a period of significant cooling between 1950 and 1985 (-0.4 °C decade$^{-1}$, P<0.01) followed by a strong warming from 1985 to 2022 (+0.7 °C decade$^{-1}$, P<0.01). Both the cooling that occurred until 1985 and the subsequent warming were most pronounced in central and southern Greenland (Figure 7a-b). In contrast, the low elevations of the northwest Greenland ice sheet underwent warming during the entire period (Figure 7a–c).

For the time period for which ANN, RACMO, HIRHAM and MAR are available (1980-2016), the ANN gives an ice-sheet-wide average $T_{10m}$ trend of +0.6 °C decade$^{-1}$ (P<0.01) , while the equivalent trends are estimated at +0.3, +0.4, and -0.1 °C decade$^{-1}$ by RACMO, HIRHAM and MAR (P≤0.01), respectively (Table 3, Figure 6a). The spatial patterns of $T_{10m}$ trends in the three RCMs (Figure 7e-g) are consistent with the ANN (Figure 7d): a more pronounced warming at a mid-elevation band around the ice sheet and a milder warming (or cooling for MAR) in the rest of the ice sheet.

Since the processes controlling $T_{10m}$ depend on the local climatic, snow and ice conditions, we also compare the evolution of $T_{10m}$ in different ice sheet regions (Figure 1): i) the bare ice area where seasonal snow melts completely and exposes underlying glacial ice at the end of summer, ii) the dry snow area where little or no melt occurs, and iii) the intermediate percolation area where a significant portion of the annual snow accumulation melts in spring and summer and percolates into the underlying firn (Figure 1a). In the bare ice area (Figure 6b), the observation-based ANN predicts stable $T_{10m}$ until the 1980s and increasing $T_{10m}$ thereafter (+0.6 °C decade$^{-1}$ over 1985-2022, P<0.01). In contrast, MAR estimates a negative trend in $T_{10m}$ temperatures over the 1950-2022 period and overestimates the $T_{10m}$ during the 1950-2000 period compared to the ANN (Figure 6b). In the bare ice area, RACMO and HIRHAM both present a $T_{10m}$ trend of +0.2 °C decade$^{-1}$ (P<0.01) over 1980-2016 period,

which is 66% smaller than the ANN trend in the ablation area for the same period. In the dry snow area (Figure 6c), there is a better agreement between the models but lower $T_{10m}$ in the 1990s in the ANN leads to a more pronounced warming trend in that area (+0.5 °C decade$^{-1}$ over 1980-2016, P<0.01) which is 40-60 % larger than warming trends predicted by RACMO and HIRHAM. MAR describes an overall cooling in the dry snow area

(Figure 6c, 7g, Table 3). In the percolation area (Figure 6d), MAR has a warm bias compared to the other models (~+4 °C on 1980-2016), but all models agree on the strong warming that occurred here since the 1980s: between +0.5 and +0.9 °C decade$^{-1}$ (all P<0.01) over 1980-2016 (Figure 6d, 7d-g, Table 3). Overall, these spatial differences average into a warm bias of MAR for the entire ice sheet and more pronounced trends for the ANN than for the RCMs (Figure 6a).

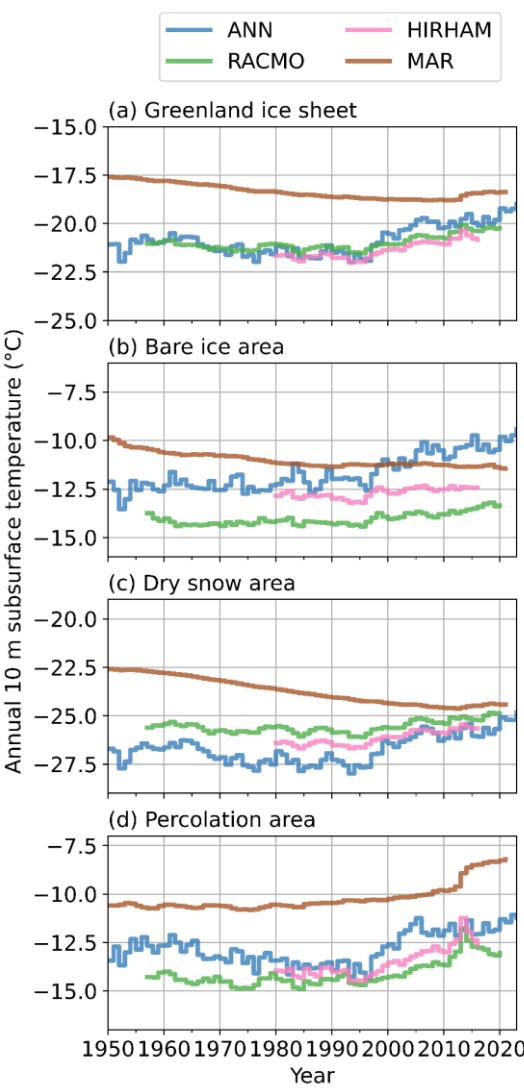


**Figure 6. Evolution of the 10 m subsurface temperature (T$_{10m}$) for all of the Greenland ice sheet (a) and in three ice sheet regions (b-d). Although all panels have the same vertical axis scaling, note the different vertical axis bounds.**

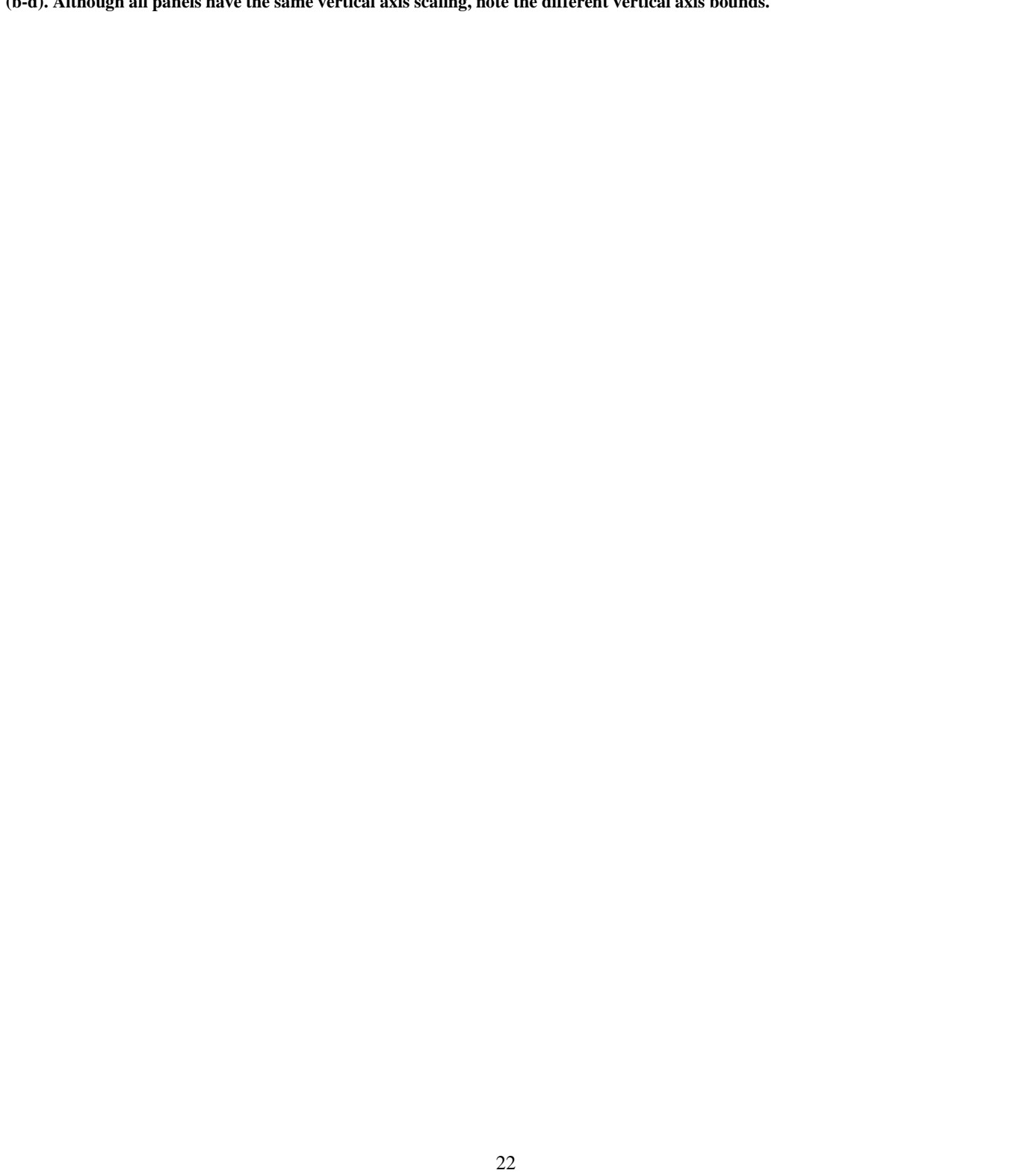

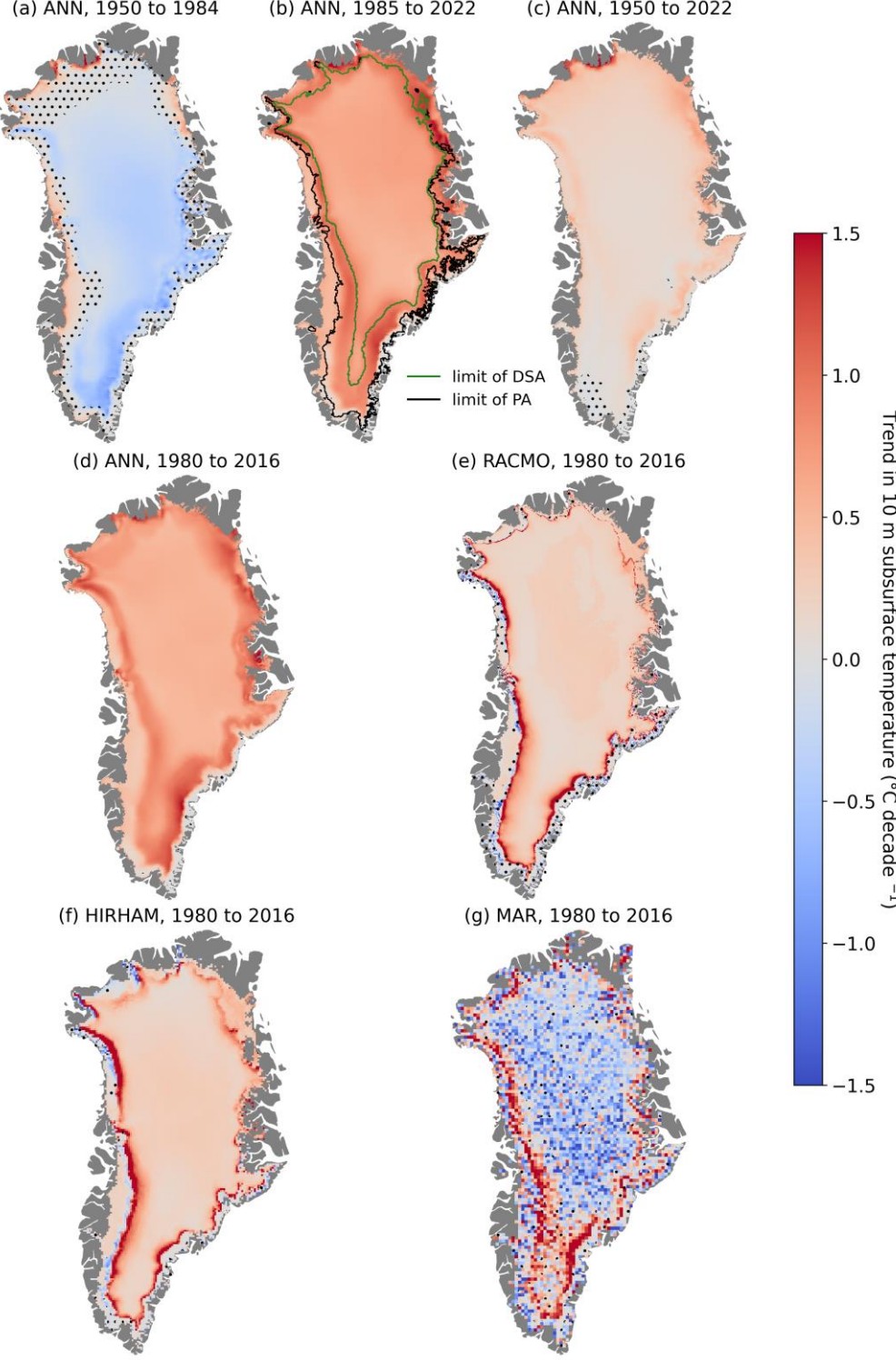


**Figure 7. Trends in 10 m subsurface temperature as determined by the ANN over the periods 1950-1984 (a), 1985-2022 (b), 1950-2022 (c) and 1980-2016 (d), and calculated by RACMO (e), HIRHAM (f) and MAR (g) over the period 1980-2016, when data from all models are available. Dotted areas indicate trends below significance level (P > 0.1). In panel b, the lower limit of the dry snow area (DSA) and of the percolation area (PA) are shown in dark green and black, respectively.**


**Table 3. Trends in 10 m subsurface temperature for different ice sheet regions and different periods. All trends are significant at a 0.1 level.**

| Model | Period | Mean $T_{10m}$ | Trend in $T_{10m}$ ($°C$ decade$^{-1}$) |
|---|---|---|---|
| *All Greenland ice sheet* | | | |
| ANN | 1950-1985 | -21.5 | -0.4 |
| ANN | 1985-2022 | -20.7 | 0.7 |
| ANN | 1950-2022 | -21.1 | 0.2 |
| ANN | 1980-2016 | -21.0 | 0.6 |
| RACMO | 1980-2016 | -21.0 | 0.3 |
| HIRHAM | 1980-2016 | -21.4 | 0.4 |
| MAR | 1980-2016 | -18.6 | -0.1 |
| *Bare ice area* | | | |
| ANN | 1950-1985 | -10.8 | 0.1 |
| ANN | 1985-2022 | -9.5 | 0.6 |
| ANN | 1950-2022 | -10.1 | 0.4 |
| ANN | 1980-2016 | -9.7 | 0.6 |
| RACMO | 1980-2016 | -14.0 | 0.2 |
| HIRHAM | 1980-2016 | -13.0 | 0.2 |
| MAR | 1980-2016 | -11.3 | -0.0 |
| *Dry snow area* | | | |
| ANN | 1950-1985 | -27.1 | -0.4 |
| ANN | 1985-2022 | -26.5 | 0.6 |
| ANN | 1950-2022 | -26.8 | 0.2 |
| ANN | 1980-2016 | -26.8 | 0.5 |
| RACMO | 1980-2016 | -25.6 | 0.2 |
| HIRHAM | 1980-2016 | -26.1 | 0.3 |
| MAR | 1980-2016 | -24.2 | -0.3 |
| *Percolation area* | | | |
| ANN | 1950-1985 | -13.5 | -0.5 |
| ANN | 1985-2022 | -12.8 | 0.9 |
| ANN | 1950-2022 | -13.1 | 0.2 |
| ANN | 1980-2016 | -13.1 | 0.8 |

| | | | |
|---|---|---|---|
| RACMO | 1980-2016 | -14.0 | 0.6 |
| HIRHAM | 1980-2016 | -13.5 | 0.6 |
| MAR | 1980-2016 | -10.1 | 0.4 |

4. **Discussion**

We compiled the largest dataset of observed subsurface temperature on the Greenland ice sheet to date and used it to train an ANN which, with snowfall and temperature from ERA5 reanalysis as input, estimates monthly grids of 10 m subsurface temperature over the entire ice sheet for the 1950-2022 period. The ANN describes a -0.4 °C decade$^{-1}$ $T_{10m}$trend during 1950-1985 (Figure 6a, Table 3) which is consistent with the negative trends in air temperatures found by Zhang et al. (2022) in the ERA5 reanalysis and RACMO RCM from the late 1950s to early 1990s. The following increase in $T_{10m}$ (+0.7 °C decade$^{-1}$) calculated by the ANN from the 1990s to 2022 is

consistent with all-year and summer air temperature increases found in weather station observations, reanalysis datasets and regional climate models (Hanna et al., 2021, Zhang et al., 2022). The ice sheet wide average $T_{10m}$ trend, +0.2 °C decade$^{-1}$ over 1950-2022, agrees with the trend in annual air temperature in ERA5 (+0.2 °C decade$^{-1}$ over 1950-2022). Additionally, the ANN estimates a strong warming of +0.9 °C decade$^{-1}$ on average, up to +1.4 °C decade$^{-1}$ locally, in the percolation area (bounded by the dark green and black lines in Figure 7b)

during the 1985-2022 period.. This localized warming of the percolation area is also calculated by the three RCMs (Figure 6d, Figure 7e-g). However, this hotspot of $T_{10m}$ increase is not found in air temperature trends (Zhang et al., 2022 Fig. 5-7). The warming of the subsurface in the percolation area stems from the increased meltwater infiltration and from the latent heat released by refreezing (e.g. Humphrey et al., 2012; Vandecrux et al., 2020a). This successful identification of areas subject to firn warming by the ANN is remarkable considering

that the ANN only learns from the $T_{10m}$ observations, and the local air temperature and snowfall history, and is not fed information on meltwater infiltration and refreezing. This indicates that the ANN successfully learns which areas are susceptible to undergo meltwater infiltration and refreezing from its training data.

The ANN model has the strength of statistical models: it fits the training data and thereby performs better than RCMs when evaluated against observations used for training (Figure 3a, 4). Yet, ANNs and statistical models have several limitations. Firstly, the ANN is greatly dependent on the distribution of the training data, and how representative that data is of the parameter space where the ANN is applied. Our methods give more weight to observations that are from underrepresented areas of the parameter space. Yet, there are still regions with

particular combinations of air temperature and snowfall where no observations are available and where the ANN extrapolates. More observations are needed from these less-visited parts of the ice sheet to further train the ANN. These new measurements could either focus on the coldest parts of the ice sheet, where our compilation currently lacks measurements (Figure 2a) or on the areas where our uncertainty is the highest, in the Southeast (Figure 3c). Secondly, the ANN is limited by the input parameters it draws on. For instance, inaccuracies in ERA5 data (as discussed in Delhasse et al., 2020; Zhang et al., 2022) for certain periods or locations will affect the performance of the ANN, as will $T_{10m}$ measurement uncertainties. Besides, only using two input parameters (air temperature and snowfall) must introduce inaccuracy through oversimplification of complex physical processes. Additionally, the relatively coarse resolution of the input grid ($0.1 \times 0.1°$) prevents the ANN from identifying local phenomena such as localized meltwater refreezing in surface deepenings and crevasses (Hills et al., 2018, Chudley et al., 2021) or in ephemeral perched aquifers (Humphrey et al., 2021, Culberg et al., 2022). Nor can our ANN model account for the exposure of ice affected by past temperature anomalies, i.e. the advection of deep ice in the ablation zone that may drive $T_{10m}$ more than surface conditions (Lüthi et al., 2015). Other widespread processes such as the penetration of short-wave radiation into the subsurface (Van den Broeke et al., 2008; Kuipers Munneke et al., 2009; Van Dalum et al., 2021), firn ventilation (Albert and Shultz, 2002) or potentially 'wind pumping' (Clarke et al., 1987) are more likely to be accounted for by the ANN because observations subject to these processes are included in the dataset. Ultimately, the ANN cannot identify the processes that are responsible for a given subsurface temperature, but it can learn which $T_{10m}$ are usually seen at various temperature and snowfall combinations.

Although the RCMs calculate subsurface temperatures in similar ways (see Section 2.3), differences arise due to their various assumptions. For example, MAR assumes that in the ablation area, the material added at the lower bound of the model column is always slightly colder than the lowermost material left in the column. This explains the decreasing trend in simulated $T_{10m}$ in Figure 6a and the inability of MAR to explain the observed $T_{10m}$ variation at the ablation sites (Figure 4c). The noise within the $T_{10m}$ trend map (Figure 7g) is also indicative of some numerical instability in this deep temperature prescription. These limitations of the model's boundary conditions have now been identified and efforts are ongoing to remediate them in the next version of MAR. It is, however, interesting to note that these biases do not significantly impact the surface mass balance simulated by MAR; different sensitivity tests were performed with the aim of improving the comparison with $T_{10m}$ and for all of them, the MAR results at the surface remained unchanged. Langen et al. (2017) showed that the simulated subsurface temperature profile in HIRHAM in the percolation area greatly depends on the formation, in the

model, of ice layers of density greater than 830 kg m$^{-3}$ that inhibit water infiltration. The formation of these high density layers in the model depends on the surface climate and subsurface model, but also on the discretization of the modeled firn column, which is currently fixed in HIRHAM (Vandecrux et al., 2020b). Recent efforts to update the HIRHAM subsurface scheme to a more flexible discretization that would preserve high density layers

was made for Antarctica (Hansen et al. 2021) but has not yet been applied to Greenland. Steger et al. (2017) found that the SNOWPACK model forced by RACMO2.3, an older version of RACMO2.3p2 evaluated here, overestimated the subsurface temperature in the high elevation areas in northwestern Greenland, while underestimating the firn temperature at lower elevations due to either insufficient meltwater generation at the surface or too shallow simulated meltwater infiltration. In that same study, RACMO2.3 in combination with both

IMAU-FDMv1.1 and SNOWPACK subsurface schemes could not accurately reproduce subsurface temperatures at some low percolation sites because the models represented them as bare ice sites. This mismatch between the simulated and actual surface type – bare ice or porous firn – makes sites at the transition between the bare ice and percolation areas, i.e., the equilibrium line, particularly challenging for all RCMs (e.g., KAN_U, Swiss Camp, KPC_U in Figure 5d-f). Switching from version 2.3 to 2.3p2, in combination with an update to IMAU-FDMv1.2

allowed RACMO to simulate KAN_U as a firn site rather than a bare ice site (Ligtenberg et al., 2018, Brils et al., 2022). The IMAU-FDM always allows meltwater infiltration, which may lead to an overestimation of $T_{10m}$ at sites where thick ice layers in the firn provide a barrier for further percolation. This was highlighted at KAN_U when driving IMAU-FDMv1.1 with surface temperature and melt rates derived from observations (Vandecrux et al., 2020b). However, the updated IMAU-FDMv1.2 forced by RACMO2.3p2 now shows a slight cold bias at

KAN_U (Figure 5d), indicating that too deep meltwater infiltration is no longer an issue at that site (Brils et al., 2022).

The subsurface temperature impacts the surface energy budget through the conductive heat flux, and thereby affects the snow and ice surface melt. Heat from a warm subsurface will be conducted to the surface when

surface temperatures are lower. And vice-versa, a colder subsurface represents a heat sink (heat will conduct down, away from the surface) and will moderate surface melt. Another consequence of the near-surface snow and firn warming is that it decreases the cold content and therefore the retention capacity of the snow and firn (Pfeffer et al., 1991; Vandecrux et al., 2020a). Meltwater retention in firn occurs when i) pore space is available, ii) this pore space can be accessed by the meltwater and iii) cold content is available to refreeze the meltwater.

Vandecrux et al. (2019) documented that the upper 10 m of the firn in the lower accumulation and percolation area lost about 20% of its pore space over the last decades, while in the dry snow area pore space remained stable

since the 1950s. Our work documents the recent subsurface warming of the ice sheet and how the upper 10 m of snow, ice and firn is brought closer to the melting point, potentially enhancing meltwater runoff in the subsequent summers. Our ANN estimates that the dry snow area average $T_{10m}$ increased from -27.3 °C over 1980-1990 to -

25.8 °C over 2010-2020. Similarly, the percolation area warmed from -13.7 °C over 1980-1990 to -11.8 °C over 2010-2020; 1.9 °C (14%) closer to the melting point. Our findings complement other work showing the changes in the Greenland ice sheet subsurface their impact on the ice sheet mass loss: for example the recent expansion of the firn aquifer area stemming, among other causes, from the loss of firn cold content (Horlings et al., 2022) or the increasing the runoff from the firn area (Tedstone and Machguth, 2022) linked to the formation of ice layers

reducing meltwater percolation and retention into the underlying firn (Machguth et al., 2016; MacFerrin et al., 2019) .

5. **Conclusion**

Using the most complete compilation of observed $T_{10m}$ on the Greenland ice sheet to date, we trained an Artificial Neural Network (ANN) to describe the spatio-temporal evolution of $T_{10m}$ during 1950-2022. We found that,

following a significant cooling between 1950 and 1985 (-0.4 °C decade$^{-1}$, P<0.1), ice sheet-wide $T_{10m}$ increased by +0.7 °C decade$^{-1}$ from 1985 to 2022 (P<0.1). Overall, the Greenland ice sheet $T_{10m}$ increased at a rate of +0.2 °C decade$^{-1}$ over the 1950-2022 period in response to increasing energy influx at the surface. Our observational dataset yielded unique and extensive constraints on the subsurface temperature simulated by three conventional regional climate models, RACMO, MAR and HIRHAM and demonstrated their mixed performance. Notably, it

revealed numerical instabilities in MAR prompting improvements in its snow module, although these $T_{10m}$ biases apparently have low impact on the SMB simulated by MAR. This work highlights the value of in-situ measurements of ice, snow and firn temperatures to better quantify the response of the Greenland ice sheet to Arctic warming and to reduce uncertainty in projections of mass loss from the Greenland ice sheet. Our evaluation shows highest ANN uncertainty in the southeast and in the lower percolation area in northern

Greenland (Figure 3). Those are regions where few observations are available (Figure 1) and consequently, any additional measurements there will help to constrain models and understand the relevant processes.

6. **Data and code availability**

The original subsurface temperature datasets are cited in Table 1 and, when available, download links to the original datasets can be found in the reference list. Some of the observational data was found in other

compilations such as Mock and Weeks (1965), McGrath et al. (2013) or Løkkegaard et al. (2023) and Mankoff et al. (2022).

The monthly $T_{10m}$ dataset is currently hosted at https://doi.org/10.22008/FK2/TURMGZ (Vandecrux, 2023a) and is also added to the 2023 release of the SUMup dataset (Vandecrux et al., 2023b). A compilation of non-interpolated, instantaneous subsurface temperatures can be requested from the authors. The $T_{10m}$ maps here are available at https://doi.org/10.22008/FK2/C24WVN (Vandecrux, 2023b) and the scripts used for the analysis are available at https://doi.org/10.5281/zenodo.8027442 (Vandecrux, 2023c).

## 7. Acknowledgments

BV designed the study, processed and compiled the observations, and conducted the analysis with the help from RSF, JEB, FC, RH, AKR. RSF. RSF, APA, JEB, FC, RH, AKR, AH, JA, PCJPS and DvA funded or organized projects measuring subsurface temperatures and provided valuable data. EB provided insights in the ANN setup and tuning. XF, PKM, MRvdB, MB, PLL, RM provided RCM outputs and their insights on their performance. BV drafted the manuscript, to which all co-authors contributed.

## 8. Competing interests

XF, RM and MRvdB. are members of the editorial board of the Cryosphere.

## 9. Acknowledgments

We salute the field parties that collected data in the field over more than 100 years, allowing us to conduct the present analysis. The PROMICE and the new GC-Net AWS are supported by the Danish Ministry for Environment, Energy and Utilities. The historical GC-Net AWS and FirnCover were supported by NASA, NSF and WSL grants. JA was supported by the Austrian Science Fund (P35388). F. Covi, Å. Rennermalm and R. Hock were supported by NSF grant no. 397516-66782. We also thank Hubertus Fischer, Wilfried Häberli, Olaf Eisen, Shin Sugiyama, Sumito Matoba and Robert Hawley for sharing their subsurface temperature data. MRvdB, PKM and MB acknowledge funding from the Netherlands Earth System Science Centre (NESSC). PLL gratefully acknowledges funding from the Aarhus University Interdisciplinary Centre for Climate Change

(iClimate, Aarhus University). AH has been supported by the DFG (Deutsche Forschungsgemeinschaft; grant no. HE7501/1-1).

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
