# Peer review of "Recent warming trends of the Greenland ice sheet documented by historical firn and ice temperature observations and machine learning"

_The Cryosphere, 2023_

## Author Comment (AC1)

Thank you for this positive review of our work and for the careful proofreading. We answer each point below in blue text .

Yours sincerely,

Baptiste Vandecrux on behalf of the co-authors
* * *
**General comments**

1) When I first read the title of the manuscript, I thought the documented warming came directly from observations. Instead, the warming trends come from output of the ANN. I understand that documenting widespread warming from observations is challenging because observations from a single location do not exist over long time periods and there is a lack of spatial coverage across the ice sheet. However, I have two recommendations to address this issue. The first would be to slightly modify the title. Maybe something along the lines of "Historical firn and ice temperature observations inform modeled warming trends across the Greenland ice sheet" (something along those lines).

The title was updated to: *Recent warming trends of the Greenland ice sheet documented by historical firn and ice temperature observations and machine learning*

My second suggestion is to actually try to evaluate how well ANN, HIRHAM, RACMO, and MAR temperatures compare to observations over time.

We have done our best to do precisely this in section 3.2 "RCM evaluation and comparison with the ANN". The evaluation of the ANN done in section 3.1 is done through the evaluation of spatial cross-validation. We also added an evaluation of the trends calculated by the ANN against observations which was also a suggestion from the other reviewer.

Perhaps creating a temperature vs. time plot (maybe breaking it into panels by dry snow, percolation, and ablation zones) and plotting each temperature value with its corresponding date. You could then add the temperature simulated by the ANN, HIRHAM, RACMO, and MAR for that same location at the same time. In the paper you mention that all models diverge when estimating past history of sites when observations are not available. While it seems like the ANN is probably the most trustworthy model, making a plot this way might help to show that the ANN can model temperatures most accurately through time across the three regions of the ice sheet.

Our figure 5 plots temperature, observed and modeled by the ANN and RCMs, versus time, at sites representative of different climate zones. It also shows the spread between models and how the ANN matches best with observations. But maybe the suggested plot was meant to display all observations from a given zone along with the model prediction (like adding observations to Figure 6). In that case we are afraid that the seasonal variation of temperature and their spread within their respective zones would make the figure hard to read. The cloud of observations from a given zone would be uncorrected from observations' sampling bias: in the percolation areas, the observations from western Greenland outweigh those from eastern Greenland and such a cloud would misrepresent the true average temperature for that climate zone.

It could also allow you to compare observational temperature trends with model trends by performing regressions on the temp vs. time plot. Not sure – this is just my initial thought here, but I'm happy to hear other suggestions if the authors have them.

Prompted by another comment from the second reviewer, we added in Section 3.1 a more systematic evaluation of the T10m trends estimated by our ANN at sites where enough observations are available:

*To evaluate the capacity of the ANN to capture the recent evolution of $T_{10m}$, we select 10 sites where more than 60 monthly values are available between 1998 and 2022 and compare the trends calculated from the ANN and the observations over the periods 1998-2010 and 1998-2022 (Table 2). These periods were chosen because of a general lack of measurements between 2011 and 2020. Trends calculated from the ANN only consider the months where observations are available. We note that due to the missing months, these trends are not reliable for general inference on the true $T_{10m}$ evolution: depending on which months are missing it might overestimate or underestimate the true $T_{10m}$ trend for these periods. The median $T_{10m}$ trends for 1998-2010 are 0.9 and 0.8 °C decade$^{-1}$ for the ANN and for the observations respectively (Table 2). For the period 1998-2022, the median $T_{10m}$ trends for 1998-2010 are 0.4 and 0.6 °C decade$^{-1}$ for the ANN and for the observations respectively (Table 2). The ANN therefore slightly overestimates the $T_{10m}$ trend during 1998-2010 and underestimates it during 1998-2022. We conclude that the ANN reproduces the magnitude of the $T_{10m}$ increase seen in observations although this aptitude varies with the location and the time period considered. From this assessment and because the ANN does not suffer temporal nor spatial gaps, the ANN appears as a suitable tool to study the trends in $T_{10m}$ over the entire Greenland ice sheet.*

*Table 2: Trends in 10 m subsurface temperature ($T_{10m}$) calculated from the ANN and observations (obs.) at 10 sites for the periods 1998-2010 and 1998-2022. ANN trends are calculated only from the months where observations are also available. The difference between the two calculated trends as well as the number of monthly values used for the calculation (N) are also given for each site.*

| | Trends in T10m (°C decade-1) | | | | | | | |
| | 1998-2010 | | | | 1998-2022 | | | |
| Site | ANN | obs. | ANN - obs. | N | ANN | obs. | ANN - obs. | N |
|---|---|---|---|---|---|---|---|---|
| NASA-SE | 1.0 | 0.7 | 0.3 | 115 | 0.4 | 0.5 | -0.1 | 171 |
| NASA-E | 0.5 | 0.5 | 0.1 | 140 | 0.6 | 0.5 | 0.0 | 270 |
| Summit | 0.4 | 1.0 | -0.6 | 133 | 0.3 | 0.6 | -0.3 | 172 |
| Tunu-N | 0.7 | 0.3 | 0.4 | 140 | 0.6 | 0.5 | 0.0 | 150 |
| South Dome | 1.4 | 0.8 | 0.5 | 97 | 0.2 | 0.5 | -0.2 | 116 |
| Saddle | 1.4 | 0.7 | 0.7 | 125 | 0.2 | 0.6 | -0.4 | 156 |
| Humboldt | 0.5 | 1.0 | -0.4 | 66 | 0.4 | 0.3 | 0.1 | 71 |
| Crawford Point 1 | 1.3 | 3.0 | -1.7 | 63 | 0.4 | 0.7 | -0.3 | 120 |
| DYE-2 | 1.2 | 0.8 | 0.4 | 139 | 0.3 | 1.1 | -0.7 | 220 |
| Swiss Camp | 0.7 | 0.8 | 0.0 | 83 | 0.3 | 1.8 | -1.5 | 172 |

 2) One interesting aspect of this paper is that it helps to show where future field efforts should be  concentrated to collect subsurface temperature measurements. I'm struck by the fact that air  temperature, snow accumulation, and temperature amplitude at observation sites are generally  pretty normally-distributed compared to the distribution of these parameters across the entire ice  sheet (Figure 2). Field researchers need to target locations that will skew the

overall temperature  and accumulation distributions. In general, sites that have lower air temperatures and snow accumulation need to be targeted for additional measurements.  However, the uncertainty of the ANN is actually highest in the southeast portion of the ice sheet,  which to my understanding is characterized by higher air temperatures and accumulation rates.  Overall, we have decent coverage of these conditions at observation sites (Figure 2). However,  we have few measurements in the specific firn hydrologic regime that produces firn aquifers. Additionally, ANN uncertainty is elevated in northern Greenland (particularly Northwest – Figure  3c). This is where we have radar observations of thick ice slabs forming (a different firn  hydrology regime from firn aquifers). My question is: is it more important to collect subsurface  temperature observations in locations that improve representativity of accumulation/air  temperature conditions or should our focus be on improving representativity of observations from  different firn hydrology regimes? If you create a map of accumulation/temperature conditions where our observations are deficient, how do they compare with where ANN uncertainty is  highest?

Thank you for this interesting reflection. We agree that Figure 2 shows that the cold and dry, high elevation areas are underrepresented in our dataset and that simultaneously Figure 3 shows highest uncertainty in the southeast. The low uncertainty (Figure 1c) and good performance of the ANN (Figure 1ab) in the cold regions show that the available observations apparently are enough to simulate subsurface temperature there. So we do not think that additional measurements are needed there, especially if continuous monitoring sites like GC-Net weather stations are maintained. However, although there are measurements in the firn aquifer region, they are still few and not covering the entire southeast. The high uncertainty in Figure 3c also reflects that the ANN depends on few observations there. Where we "should focus" or where "observations are deficient" depends on our objective. If we aim at documenting the Greenland ice sheet as a whole, then the extensive high elevation plateau is of high importance. If we aim at understanding processes or at reducing the uncertainty in less representative regions like the firn aquifer region, then that is where we need to collect more measurements. We added to our discussion:

*More observations are needed from these less-visited parts of the ice sheet to further train the ANN. These new measurements could either focus on the coldest parts of the ice sheet, where our compilation currently lacks measurements (Figure 2a) or on the areas where our uncertainty is the highest, in the Southeast (Figure 3c).*

and to our conclusion:

Our evaluation shows highest ANN uncertainty in the southeast and in the lower percolation area in northern Greenland (Figure 3). Those are regions where few observations are available (Figure 1) and consequently, any additional measurements there will help to constrain models and understand the relevant processes.

**Specific comments:**

1) This comment may just be a lack of my detailed understanding of ANNs. You state on line 132- 133 that "ANNs have proven their ability to capture these non-linearities and interactions  between input parameters in numerous glaciological and meteorological applications". However,  on lines 213-214 you state that you use a rectified linear unit activation function. Isn't it the  activation function that makes ANNs either linear or nonlinear? So although you are attempting to  capture nonlinearities, are you not just using a linear ANN? Again, this question is probably due  to me not fully understanding the structure of ANNs, but maybe you could add a sentence that  describes how the ANN is nonlinear.

We see that it can be counter intuitive that the ANN is a non-linear model made up of many piecewise linear functions. First, it needs to be said that the activation function we use is a piecewise linear function that gives 0 if the input is below 0, f(x)=x if input x is above 0, and is not linear strictly speaking. But more importantly, it is the interactions between the neurons that makes ANNs able to reproduce highly nonlinear functions. Taking for example an input x and a target function f=exp(x), for an ANN of one layer and three neurons, f is approximated by a combination of three piecewise functions, meaning a line with three breaks and four segments. If we add another layer to the ANN, each neuron on the second layer can make its own weighted sum of the output of the three previous neurons, apply an offset (called bias) and apply their activation function to the result. These weights and biases are as many degrees of freedom that allow to tune the network to fit even non-linear functions.

2) I think that it is important to mention that in Figures 3 and 5 that the observations should  generally be well-reproduced by the ANN, because the model has been trained explicitly "seeing"  these data, which is probably why, especially in Figure 5, the ANN appears to outperform other  models over periods where observational data exist.

When reaching Figure 5, the reader is aware that the ANN is trained on the observations. This is one of the strengths of the ANN, and we describe at length in Section 2.2 and 2.3 how the ANN and RCM are of different nature. In our result section, we  did our best to be nuanced in our analysis and avoid one-to-one comparison of the ANN with the RCM for exactly that reason. That is why for instance, the scatter plots of the ANN (Figure 3) are not presented along with similar plots from the RCM (Figure 4). We therefore prefer to avoid further repetition of how the ANN is built while describing its performance. Last but not least, the presented RCMs are regularly evaluated and updated to match the available observations (see f.e. Langen et al., 2017, Vandecrux et al., 2022, Brils et al., 2022) and therefore also "see" the observations, just in a different and less direct way than the ANN.

**Technical corrections/comments:**

These are some typos that I have caught or comments that are meant to improve clarity of the text.

**L1:** Should "firn" replace "snow" in the title, since by 10 m you are observing firn temperatures rather  than snow?

We updated the title to "*Recent warming trends of the Greenland ice sheet documented by historical firn and ice temperature observations and machine learning*"

**L22:** Specify what about "melting". Melt production? Melt rates? Melt extent, duration?

We added *in intensity and extent*

**L42:** change "undergoing" to "experiencing" or "exposed to"?

We changed to *exposed to.* Thanks.

**L51:** Perhaps add "and grain coarsening" after "liquid water within snow"? You can cite Nolin and  Stroeve (1997) for this.
Added, thanks.

**L53:** Is Humphrey et al. (2012) not an appropriate reference here?
Added, thanks.

**L57:** note that the "retaining meltwater" is retention through refreezing

We updated to *refreezing and retaining meltwater*.

**L58:** Maybe caveat this statement – the degree to which cryo-hydrologic warming will increase dynamic  mass loss depends on the glaciological setting (e.g., Poinar et al. 2017).

Thank you for pointing out this study. We rephrased to:

*Subsurface warming could also [...]and increase the ice viscosity (Phillips et al., 2010, 2013, Colgan et al., 2015) although with limited impact on dynamic mass loss (Poinar et al., 2017).*

**L63:** Just noting the inconsistency. You state 4500 in the abstract and here you say 4600.

Thank you for spotting this. These numbers were actually both inaccurate and we updated them to *4612* for the number of observations in our compilation and *4597* the number of observations being used to train our ANN. We

added details about these two numbers in Section 2.1:

*Among these 4612 $T_{10m}$ observations, 15 measurements are either outside of the current ice sheet extent as defined by the GIMP ice sheet delineation (Howat et al., 2014) or outside of the 1950-2022 period we consider for our $T_{10m}$ reconstruction. There are therefore 4597 $T_{10m}$ observations in our compilation that can be used for the reconstruction of $T_{10m}$ on the ice sheet between 1950 and 2022.*

**L65:** I know you state this in the methods section, but it may be worth just adding a sentence as to why you choose an ANN over other machine learning models in the introduction.

We would like to keep the introduction focused on the motivation of the study and keep the technical details for the Methods section. Adding a sentence about why we choose it over other methods would require that we define, also in the introduction, what the ANN is, how it is being used and eventually why we prefer it over other techniques (Section 2.2). For the sake of concision, we keep model description to the methods.

**L71-72:** I'm not entirely sure what you mean by "We put special emphasis on T10m magnitude and trends in various areas of the ice sheet". Maybe something like: "We evaluate the differences between observed and modeled T10m across the entire ice sheet and specifically in the ablation area, and we evaluate modeled temperature trends in bare ice regions, the percolation zone, and the dry snow zone of the ice sheet."

Thank you for the suggestion. We updated to:

*Using our observational dataset of subsurface temperature as well as our ANN, we evaluate three regional climate models (RCMs) widely used to estimate the surface mass balance of the Greenland ice sheet [...]. We then evaluate the ANN and RCMs' T10m magnitudes and trends in the bare ice, percolation, and dry snow areas of the ice sheet. Lastly, we discuss the impact of this subsurface warming on the ice sheet mass balance processes.*

**L78:** change "thermistor strings," to "thermistor strings:"

Updated, thanks.

**L81:** change "was operating" to "operated"

Updated, thanks.

**L81:** change "and equipped" to "and was equipped"

Updated, thanks.

**L83:** change "and active" to "and was active"

Updated, thanks.

**L84:** change "along" to "over"

Updated, thanks.

**L85:** delete "yet"

Updated, thanks.

**L88:** add the sensor spacing here.

Updated, thanks.

**L91:** change "was" to "were"

Updated, thanks.

**L95-96:** Maybe clearer to write "Using firn temperature observations reported by Samimi et al. (2012) and Heilig et al. (2018) at Dye-2 as a reference..."

Updated, thanks.

**L105:** change "was" to "were"

Updated, thanks.

**L106-107:** Why can Benson not be used – too shallow? Also, why do you include observations that were not collected on the Greenland ice sheet?

Thank you for pointing this out. Those measurements fell outside of the GIMP ice mask (Howat et al., 2014) so we meant that we do not use them for the T10m reconstruction. They are still in the compilation of T10m measurements. We moved this part to a different paragraph and rephrased it to:

*Among these 4612 $T_{10m}$ observations, 15 measurements are either outside of the current ice sheet extent as defined by the GIMP ice sheet delineation (Howat et al., 2014) or outside of the 1950-2022 period we consider for our $T_{10m}$ reconstruction. There are therefore 4597 $T_{10m}$ observations in our compilation that can be used for the reconstruction of $T_{10m}$ on the ice sheet between 1950 and 2022.*

**L148-149:** A little clearer to say "Additionally, for each grid cell and monthly time step we calculate…"

Updated, thanks.

**L152-153:** May be clearer to write: "Lastly, to assist the ANN in capturing the annual periodicity, we input the cosine of the month (assigning 1 in January and -1 in July).

Updated, thanks.

**L157:** change "ANN" to "ANNs"?

Updated, thanks.
**L163:** change "will be" to "is"

Updated, thanks.

**L181:** "and inversely" is a little unclear. May be better to just say this explicitly… "while the weight will be less than one if the observation histogram is greater than target histogram."

Updated, thanks.

**L222:** change "each the" to "each of the"

Updated, thanks.

**L305:** change "evaluate directly" to "directly evaluate"

Updated, thanks.

**L314:** change "is" to "are"

Updated, thanks.

**L335-336:** Should this sentence reference Figure 5?

This paragraph is dedicated to Figure 4 which supports the issue MAR has in the entire ablation area. Figure 5 indeed illustrates this statement at a few ablation sites (but not for the entire ablation area) and we would like to wait until Figure 5 is properly introduced in the next paragraph, before we reference it.

**Figure 5 caption:** change "y-axis" to "y-axes"

Updated, thanks.

**L386:** Isn't MAR overestimating between 1950-2000 compared to the ANN?

Well spotted, thank you.

**L412:** Maybe instead of "decrease" just "trend" or "change" since the negative sign is already included

Updated, thanks.

**L425-426:** Clearer to write "This indicates that the ANN successfully learns which areas are susceptible to undergo meltwater infiltration and refreezing from the ANNs training data"

Updated, thanks.

**L429:** change "ANN" to "ANNs"?

Updated, thanks.

**L441:** change "Neither" to "Nor"?

Updated, thanks.

**L488:** note that "retention capacity" is retention capacity from refreezing rather than storage in perennial or ephemeral firn aquifers.

In the following sentence, we elaborate on retention through refreezing so we think that this part is clear.

**L495:** note that the averages are from the ANN.

We now mention that they are from the ANN.

**References cited in review:**

Nolin, A. W., & Stroeve, J. (1997). The changing albedo of the Greenland ice sheet: Implications for climate modeling. *Annals of Glaciology*, *25*, 51-57. https://doi.org/10.3189/S0260305500013793

Poinar, K., Joughin, I., LENAERTS, J. T., & Van Den Broeke, M. R. (2017). Englacial latent-heat transfer has limited influence on seaward ice flux in western Greenland. *Journal of Glaciology*, *63*(237), 1-16. https://doi.org/10.1017/jog.2016.103

Brils, M., Kuipers Munneke, P., van de Berg, W. J., and van den Broeke, M.: Improved representation of the contemporary Greenland ice sheet firn layer by IMAU-FDM v1.2G, Geosci. Model Dev., 15, 7121–7138, https://doi.org/10.5194/gmd-15-7121-2022, 2022.

Howat, I. M., Negrete, A., and Smith, B. E.: The Greenland Ice Mapping Project (GIMP) land classification and surface elevation data sets, The Cryosphere, 8, 1509–1518, https://doi.org/10.5194/tc-8-1509-2014, 2014.

Langen, P. L., Fausto, R. S., Vandecrux, B., Mottram, R. H., & Box, J. E. (2017). Liquid water flow and retention on the Greenland ice sheet in the regional climate model HIRHAM5: Local and large-scale impacts. *Frontiers in Earth Science*, *4*, 110. https://doi.org/10.3389/feart.2016.00110

Vandecrux, B., Mottram, R., Langen, P. L., Fausto, R. S., Olesen, M., Stevens, C. M., Verjans, V., Leeson, A., Ligtenberg, S., Kuipers Munneke, P., Marchenko, S., van Pelt, W., Meyer, C. R., Simonsen, S. B., Heilig, A., Samimi, S., Marshall, S., Machguth, H., MacFerrin, M., Niwano, M., Miller, O., Voss, C. I., and Box, J. E.: The firn meltwater Retention Model Intercomparison Project (RetMIP): evaluation of nine firn models at four weather station sites on the Greenland ice sheet, The Cryosphere, 14, 3785–3810, https://doi.org/10.5194/tc-14-3785-2020, 2020.

---

## Author Comment (AC2)

Thank you for the positive review and constructive suggestions. Please find our response to your comments below in blue text.

Yours sincerely,

Baptiste Vandecrux on behalf of the co-authors
* * *
Major comments

- I think the Introduction should provide more context to the reader of the relevance of the 10m firn temperature. Why not surface temperature, or 20m temperature? It's not that I question it, but I think it needs to be discussed in the introduction. I assume that it is related to the fact that it sometimes is considered the bottom of the near surface layer, where seasonal variations are more or less fully dissipated (see for example Fig. 1.3 in Ligtenberg, 2014). Keenan et al. (2021) also simulates only the uppermost 10m of firn, arguing that at depth, the firn temperature equilibrates with annual average air temperature (in the absence of melt), seasonal fluctuations in temperature have dissipated and that seasonal variability in accumulation (for example density fluctuations) also has been diffused over the firn layer when it reaches 10m depth. I assume that authors have this kind of reasoning in mind when deciding to focus on 10m temperatures, but maybe data availability also plays a role. In any case, I think this should be briefly discussed in the introduction.

Thank you for pointing this out. We added a mention of the use of the 10m as a standard depth for our analysis:

*Of all depths measured, we here focus on measurements at, or close to, the 10 m depth. The temperature at this depth has been shown to be less affected by seasonal temperature variation and more representative of the long-term temperature and snowfall history at the surface (McGrath et al., 2013, Kjær et al., 2021). This makes it a convenient standard depth to compare temperatures from different periods and different sites.*

- I do get the impression that the trends in the machine learning model might be overestimated. This is hard to really conclude from the presented material, but looking in Fig. 3e for example, the trend between 1998 and 2009 seems stronger in the ANN than in the observations. [...] Is it possible to construct a figure with the trend in observations vs the trend in the ANN reconstructed temperature for time periods where there is overlap? I'm curious to see if such a scatter plot would confirm my impression of an overestimation of the trends in the ANN. Note that depending on the outcome of the correlation between observed and ANN temperature trends, the statements in the Conclusions may be weakened a bit by adding that the reported temperature trends were derived from the ANN. The statements currently are formulated in a rather absolute sense.

Thank you for this careful examination of our results. We added the following paragraph in section 3.1 and we hope it helps to clarify that there is no systematic overestimation of the T10m trends in the ANN:

*To evaluate the capacity of the ANN to capture the recent evolution of $T_{10m}$, we select 10 sites where more than 60 monthly values are available between 1998 and 2022 and compare the trends calculated from the ANN and the observations over the periods 1998-2010 and 1998-2022 (Table 2). These periods were chosen because of a general lack of measurements between 2011 and 2020. Trends calculated from the ANN only consider the months where observations are available. We note that due to the missing months, these trends are not reliable for general inference on the true $T_{10m}$ evolution: depending on which months are missing it might overestimate or underestimate the true $T_{10m}$ trend for these periods. The median $T_{10m}$ trends for 1998-2010 are 0.9 and 0.8 °C decade$^{-1}$ for the ANN and for the observations respectively (Table 2). For the period 1998-2022, the median $T_{10m}$ trends for 1998-2010 are 0.4 and 0.6 °C decade$^{-1}$ for the ANN and for the observations respectively (Table 2). The ANN therefore slightly overestimates the $T_{10m}$ trend during 1998-2010 and underestimates it during 1998-2022. We conclude that the ANN reproduces the magnitude of the $T_{10m}$ increase seen in observations although this aptitude varies with the location and the time period considered. From this assessment and*

*because the ANN does not suffer temporal nor spatial gaps, the ANN appears as a suitable tool to study the trends in $T_{10m}$ over the entire Greenland ice sheet.*

*Table 2: Trends in 10 m subsurface temperature ($T_{10m}$) calculated from the ANN and observations (obs.) at 10 sites for the periods 1998-2010 and 1998-2022. ANN trends are calculated only from the months where observations are also available. The difference between the two calculated trends as well as the number of monthly values used for the calculation (N) are also given for each site.*

| Site | Trends in T10m (°C decade-1) | | | | | | | |
|------|------|------|-----------|-----|------|------|-----------|-----|
| | **1998-2010** | | | | **1998-2022** | | | |
| | ANN | obs. | ANN - obs. | N | ANN | obs. | ANN - obs. | N |
| NASA-SE | 1.0 | 0.7 | 0.3 | 115 | 0.4 | 0.5 | -0.1 | 171 |
| NASA-E | 0.5 | 0.5 | 0.1 | 140 | 0.6 | 0.5 | 0.0 | 270 |
| Summit | 0.4 | 1.0 | -0.6 | 133 | 0.3 | 0.6 | -0.3 | 172 |
| Tunu-N | 0.7 | 0.3 | 0.4 | 140 | 0.6 | 0.5 | 0.0 | 150 |
| South Dome | 1.4 | 0.8 | 0.5 | 97 | 0.2 | 0.5 | -0.2 | 116 |
| Saddle | 1.4 | 0.7 | 0.7 | 125 | 0.2 | 0.6 | -0.4 | 156 |
| Humboldt | 0.5 | 1.0 | -0.4 | 66 | 0.4 | 0.3 | 0.1 | 71 |
| Crawford Point 1 | 1.3 | 3.0 | -1.7 | 63 | 0.4 | 0.7 | -0.3 | 120 |
| DYE-2 | 1.2 | 0.8 | 0.4 | 139 | 0.3 | 1.1 | -0.7 | 220 |
| Swiss Camp | 0.7 | 0.8 | 0.0 | 83 | 0.3 | 1.8 | -1.5 | 172 |

Generally, the seasonal fluctuations seem to be stronger in the ANN, for example in Fig. 3g. So could it be that the ANN is a little bit oversensitive to warming and cooling, whether it is seasonal, or a long term trend?

Indeed,the monthly variations of T10m produced by our ANN model are more pronounced than in observations at KPC_U (Figure 3g). We attribute the seasonal variations in the ANN to the use of the month as input to our ANN to encourage it to capture the variations in observed T10m (Figure 3g). We assume that the ANN made use of this input to fit the seasonal variations seen in observed T10m at other sites, but as a result of this fitting process, overestimates the seasonal variations at KPC_U (Figure 3g). So we believe this issue is distinct from whether our ANN captures long term trends in T10m (which has now been assessed).

- I'm not very familiar with machine learning techniques, but I'm wondering if more substantiation of the chosen features can be provided. Often, it is reported on the relative feature importance after training the ANN, such that it can be identified if the chosen features contributed significantly to the ANN. This seems to be missing a bit in the manuscript.

In section 2.2.1., we provide a description of how, through the surface energy and mass balance, air temperature and snowfall are drivers of T10m. We then describe how we calculate our inputs: the average air temperature and snowfall to provide the ANN the long-term conditions at a given time and place, the previous years' annual temperatures/snowfall and the previous year's air temperature amplitude to provide the ANN with the recent extremes and year-to-year variability, and eventually the month's cosine to represent seasonality. So we think that input variables are currently described and motivated.

The analysis of the inputs' relative importance can help identify, among several candidates, what are the main drivers of a given variable or process (in our case $T_{10m}$). This is not exactly our aim, since we already described these interactions, and that we aim at describing the recent trends in $T_{10m}$. Yet, prompted by this comment, we have applied the SHapley Additive exPlanations (SHAP) analysis (Lundberg and Lee, 2017) using the SHAP python package (https://shap.readthedocs.io). The SHAP analysis uses game theory to define the contribution of different inputs on an outcome. It is model-agnostic and is conducted on our best model (which trains on the entire compilation of $T_{10m}$ and is conducted on these same observation locations. Figure R2a presents the mean SHAP value, or impact on model output (in positive degrees contributed to the predicted $T_{10m}$), of each input variable. However these inputs are highly correlated (e.g. average temperature and snowfall are the results of the past years' annual values) and in these conditions SHAP results can be misleading: the importance of a general driver, such as snowfall, can be split into multiple variables and lead to the apparent low importance of the snowfall inputs in Figure R2a. The SHAP toolbox allows calculating the importance of groups of inputs, which we use by grouping all temperature-dependent inputs and all snowfall-dependent inputs (Figure R2b). This analysis shows that the air temperature inputs control the majority of the variability with impacts up to + 30 °C on the output, followed by the snowfall-dependent inputs (with impacts up to +10 °C on the output) and eventually the month's cosine that is responsible for a seasonal variation ranging around 1-2 °C. Note that because of the co-variation of temperature and snowfall, the individual impact of these two variables cannot be completely separated. This order of importance is in agreement with our understanding of the processes controlling $T_{10m}$ and is therefore not surprising. It confirms that the ANN works as it should and for conciseness we do not wish to add it to the main text.

[Figure]

**Figure R2.** SHAP analysis of the input variables taken individually (a) and grouped based on whether they depend on air temperature ($T_{2m}$) or snowfall (SF) (b). The inputs are ranked from most important to least important. The SHAP values are the impact of a given variable (or group of) on the output ($T_{10m}$).

Minor comments

- The title could be considered slightly misleading, suggesting that the trend is derived from the observations only. I think the title should reflect that the trends were derived from machine learning on historical data.

*The title was updated to: Recent warming trends of the Greenland ice sheet documented by historical firn and ice temperature observations and machine learning*

- Abstract, L25: here, I also recommend to briefly mention in one sentence why T_10m is relevant

We added to the abstract:

*The temperature of the ice sheet subsurface has been used as an indicator of the thermal state of the ice sheet's surface.*

The depth of 10 m, it is now justified in the introduction:

*Of all depths measured, we here focus on measurements at, or close to, the 10 m depth. The temperature at this depth has been shown to be less affected by seasonal temperature variation and more representative of the long-term temperature and snowfall history at the surface (McGrath et al., 2013, Kjær et al., 2021). This makes it a convenient standard depth to compare temperatures from different periods and different sites.*

- Abstract, L27: here, briefly introduce why you want to train the ANN. I understood that it is because the measurements vary in spatial and temporal coverage, and in order to create a unified dataset, you constructed the ANN?

We added "*point observation*" to this sentence:

*We train an Artificial Neural Network model (ANN) on our compilation of point observations [...] and use it to reconstruct $T_{10m}$ over the entire Greenland ice sheet for the period 1950-2022.*

We hope that this clarifies the interest of the ANN: going from point observations to a gap-free Greenland-wide temperature reconstruction spanning 70+ years.

- L76: Note that the abstract mentions 4500 measurements, L63 mentions 4600, and here 4615 is mentioned.

This was corrected, thank you.

- L77: Just for completeness, I would add both unpublished datasets to the table, listing them as "Unpublished" in the first column, or by making a data citation for these two unpublished datasets and cite those, after putting them on a repository, if that's allowed.

These data appear as "GC-Net unpublished" in Table 1. In the absence of better description elsewhere we would consider this article to be a suitable peer-reviewed reference for these data. Regarding their availability: their monthly average $T_{10m}$ is released along with all the other observations from our compilation, in a dedicated repository (https://doi.org/10.22008/FK2/TURMGZ). The release of the raw data was not planned at this point, because it was not necessary for the present study and we are still going through unpublished data recovered from the historical GC-Net files. The raw data will be duly released when ready.

- L81: What does UUB stand for?

At this point, only this description of the thermistor could be recovered. Trying to look it up online, I can see that other systems have been relying on so-called UUB thermistors (O'Connor et al., 1987; Mittaz et al., 2000; Hoelze et al., 2020), some of them manufactured by Fenwal. This information, although partial, is a good starting point for readers that are most curious about the instrumentation.

- L106: As I read "the latter", it only concerns the two measurements from Benson. But I assume that also the aforementioned 5 measurements from Koch also couldn't be used, since they are outside the current ice sheet extent? If not, then I don't understand this sentence.

Indeed the original sentence was unclear. We moved it to a different paragraph and reformulated to:

*Among these 4612 $T_{10m}$ observations, 15 measurements are either outside of the current ice sheet extent as defined by the GIMP ice sheet delineation (Howat et al., 2014) or outside of the 1950-2022 period we consider for our $T_{10m}$ reconstruction. There are therefore 4597 $T_{10m}$ observations in our compilation that can be used for the reconstruction of $T_{10m}$ on the ice sheet between 1950 and 2022.*

- Table 1, I think table caption should specify what non-bold numbers indicate.

We now use "*" and a footnote "*monthly mean values derived from the original measurements*".

- Section 2.2: I would recommend starting this section with a motivation for what the goal of the ANN is. See my similar comment for the Abstract.

We added:

*Point observations of $T_{10m}$ only give a partial description of the subsurface temperature: they are discontinuous in space and time. To describe the evolution of $T_{10m}$ over the entire ice sheet and over the last decades, one can train a machine learning model that links $T_{10m}$ to an input dataset which is itself continuous in space and time and assumed to drive changes in $T_{10m}$. Once the relationship between input and T10m is learned by the algorithm, the algorithm can be driven by the entire input dataset to reconstruct the $T_{10m}$ even at places where no observations are available.*

- L131: How deep is "subsurface" in this case? As far as I know, 10m temperature seems to be considered to equilibrate with annual average air or surface temperature pretty well (see major comment). In that case, this argument won't really hold? Obviously, the results presented do show that seasonal temperature fluctuations are still present, but just strongly dampened.

We have rephrased this sentence to make clear that "annual average air or surface temperature" are not the same:

*On the other hand, during periods of minimal or no melting (wintertime or nighttime in the summer), the radiative imbalance at the surface and the presence of a near-surface atmospheric temperature inversion can cause the surface temperatures, and through conduction the $T_{10m}$, to be several degrees lower than the near-surface air temperature (e.g. Miller et al., 2017, Steffen and Box, 2001).*

Note that we moved this paragraph to section 2.2.1. as it describes the motivations of picking air temperature and snowfall as input for our ANN.

- L131: Also provide an example of snowfall and its impact on firn temperature (contrasting high vs low accumulation areas for example).

Thanks for spotting this omission. We added:

*Additionally, snowfall affects the subsurface temperature in several ways. In the ablation area, the seasonal snowpack insulates the underlying ice. In the accumulation area, snow accumulated at the surface is, after some time, advected to greater depth, where it can act as either a heat source or sink depending on its temperature at time of deposition.*

Note that we moved this paragraph to section 2.2.1. as it describes the motivations of picking air temperature and snowfall as input for our ANN.

- Section 2.2.1: I think this section should provide a reference or other argumentation that ERA5 is reliable for Greenland.

Thank you for the suggestion. We added:

*Delhasse et al. (2020) showed that daily ERA5 near-surface air temperatures compare well with measurements from ice-sheet weather stations (mean bias of 0.01 °C, root mean square error of 3.05 °C). Loeb et al. (2022) found that ERA5's precipitation had the best performance out of three evaluated reanalysis datasets against weather station observations in the Canadian Arctic and in Greenland. Using airborne radar measurements of snow accumulation, Ryan et al. (2020) found that ERA5's annual snowfall in Greenland was comparable to estimates from state-of-the-art RCMs and outperformed satellite estimations.*

- L151: "changes in long term changes" I guess this must be simply: "long term changes", since otherwise it would be acceleration or deceleration of temperature trends, and that's not what's meant here I think.

Thank you for spotting this.

- Section 2.2.2: So were the weights determined time-dependently? With the temporal variability of the availability of the observations, as indicated by the histogram in Fig. 1, it may have been necessary, since during different climatological periods, the spatial coverage also varied.

I can understand that this paragraph is dense. We added an illustration that hopefully makes it more understandable:

*As an illustration, let us consider a $T_{10m}$ observation from a location that has $T_{2m\ 10yr}$ = -28°C. Figure 2a indicates that only ~10% of our observation sites have such an average temperature, compared to ~23% of the ice sheet pixels in ERA5, i.e., this sample comes from an under-represented temperature range. Following our procedure, we allocate to this observation $w_{obs}(p_1)$ = 0.23/0.1 = 2.3 to increase its final weight $w_{obs}$, which also considers the observation's representativity with regard to $SF_{10y}$ and $T_{2m,\ amp}$. Inversely, 25 % of our observation locations have  = -18°C while only 10% of the ice sheet (according to ERA5) has such average temperature (Figure 2a). Consequently, an observation having such  will receive a $w_{obs}(p_1)$ = 0.1/0.25 = 0.4 and will weigh less in the training of our ANN.*

So the weights were derived only based on the representativity of an observation's average 2m air temperature, annual air temperature amplitude and snowfall, independently of the time of collection. An old measurement, taken at a location where the climate is comparable to a contemporary measurement site, will not be given higher weight. Inversely, a recent measurement from a seldom visited climate zone, will be given a higher weight although it comes from a data-rich time period.

- Was it actually necessary to apply the weights?

We mention that "*the representativity of the training dataset compared to the target area is critical for the robustness of any machine learning model*". In that sense the weighing of observations is a way to make our training dataset more representative of the conditions found on the ice sheet: we decrease the importance of T10m observations taken from over-represented conditions, and increase the importance of observations from under-represented conditions. This is not mandatory, but a best practice.

- Would the ANN have performed much worse without them?

Here "performance" depends on our aim. If our objective is to build an ANN that reproduces our skewed, clustered observation dataset, then training the ANN on unweighted data will give the best statistics. If our objective is that the ANN gives an accurate estimate for T10m, even in data-scarce areas, then, giving higher weights to observations from these regions will force the ANN to match these observations, even though they don't represent a large fraction of the training data. The weighting will therefore decrease the visible performance, meaning increase the ME and RMSE when calculated on the skewed, clustered compilation of observations, but as a counterpart, will decrease the risk for overfitting this unbalanced training set.

- L256: Would the validation dataset also not be important to discuss to provide uncertainty bounds on the ANN?

*We are unsure what is meant by "validation dataset". We provide two bounds for the ANN uncertainty: a lower bound from the performance of our best ANN, trained on all data, and evaluated on that same data (Figure 3a). We then provide an upper bound, training 10 cross validation models that each ignore data from a given region, and then evaluating them on their unseen data (Figure 3b). We also use the spread between these cross validation models to give a spatio-temporal estimation of the uncertainty (Figure 3c).*

- L282: "HIRHAM, the use" is grammatically not a correct sentence.

*Thank you for spotting this. There was a missing word:*

*" In HIRHAM, the use of ..."*

- L307: Note that the referenced figure (Fig. 3), but also Fig. 4 separate "All sites" vs "Ablation sites", while text mentions "Dry sites" vs "Ablation sites". I guess it makes more sense to separate dry vs ablation, so I assume this is what has been done in the figures? Please make consistent.

*We removed this sentence because it was indeed strangely formulated. In the beginning of this paragraph we discuss the "all sites" statistics to give general bounds to the ANN performance. In the next section, we evaluate the RCMs and show that they are challenged at the ablation sites (Figure 4). Then the "ablation sites" statistics from Figure 3 can be mentioned explicitly:*

*The ANN, although of a different nature, gives better statistics at these ablation sites with a MD of 0.2 °C and a RMSD of 2.9 °C, even when calculated from our cross validation models' unseen data (Figure 3b).*

- L393: I suggest writing: "that occurred here".

*Updated, thank you.*

- Fig. 7: Why is the dotted area only in panel (a) and (c)? Does this mean that all other trends shown in all other subpanels of fig. 7 are significant? Maybe mention then in the caption that all modeled trends test significant.

*All panels have some insignificant trends in the ablation area. Only, for panels b and d, these areas are so narrow that they are barely visible. Therefore, I am reluctant to write that* all *trends are significant in the caption. The reader can come to the appropriate conclusion that* most *of the trends in panels b and d are significant.*

- L418: What is meant by "upper percolation area"? According to Table 2, the trend from ANN over 1985-2022 for the full percolation area is +0.9, while here, it is stated that the trend in the upper percolation area is +0.9. I'm not really sure I comprehend this. Is "upper" like high in elevation, or rather the northern parts on the ice sheet? I find it all a bit confusing. Similarly, the phrasing "this localized warming" is confusing, because the full percolation area warms with the same rate as this "upper percolation area", so is the warming then really "localized"?

*Indeed this part was not clear. We now describe the warming in the percolation area and added the boundary of the dry snow area and percolation area to Figure 7b for clarity:*

*Additionally, the ANN estimates a strong warming of +0.9 °C decade⁻¹ on average, up to +1.4 ° C decade⁻¹ locally, in the percolation area (bounded by the dark green and black lines in Figure 7b) during the 1985-2022 period.*

[Figure]

*Figure 7. Trends in 10 m subsurface temperature as determined by the ANN over the periods 1950-1984 (a), 1985 to 2022 (b), 1950 to 2022 (c) and 1980 to 2016 (d), and simulated by the RACMO (e), HIRHAM (f) and MAR (g) regional climate models over the period 1980-2016, when data from all models are available. Dotted areas indicate trends below significance level (P > 0.1). In panel b, the lower limit of the dry snow area (DSA) and of the percolation area (PA) are illustrated in dark green and black, respectively.*

References used in the review

- Ligtenberg, S.R.M. (2014) The present and future state of the Antarctic Firn layer. PhD Thesis https://dspace.library.uu.nl/bitstream/handle/1874/291634/Ligtenberg.pdf?sequence=1&isAllowed=y
- Keenan, E., Wever, N., Dattler, M., Lenaerts, J. T. M., Medley, B., Kuipers Munneke, P., and Reijmer, C.: Physics-based SNOWPACK model improves representation of near-surface Antarctic snow and firn density, The Cryosphere, 15, 1065–1085, https://doi.org/10.5194/tc-15-1065-2021, 2021.

**References:**

Breiman, Leo."Random Forests." Machine Learning 45 (1). Springer: 5-32 (2001).

Delhasse, A., Kittel, C., Amory, C., Hofer, S., van As, D., S. Fausto, R., and Fettweis, X.: Brief communication: Evaluation of the near-surface climate in ERA5 over the Greenland Ice Sheet, The Cryosphere, 14, 957–965, https://doi.org/10.5194/tc-14-957-2020, 2020.

Hoelzle, M., Hauck, C., Noetzli, J., Pellet, C., and Scherler, M.: Long-term energy balance measurements at three different mountain permafrost sites in the Swiss Alps, EGU General Assembly 2020, Online, 4–8 May 2020, EGU2020-8076, https://doi.org/10.5194/egusphere-egu2020-8076, PDF, 2020.

Howat, I. M., Negrete, A., and Smith, B. E.: The Greenland Ice Mapping Project (GIMP) land classification and surface elevation data sets, The Cryosphere, 8, 1509–1518, https://doi.org/10.5194/tc-8-1509-2014, 2014.

Lundberg, Scott M., and Su-In Lee. "A unified approach to interpreting model predictions." Advances in Neural Information Processing Systems (2017).

Mittaz, C., Hoelzle, M., & Haeberli, W.: First results and interpretation of energy-flux measurements over Alpine permafrost. *Annals of Glaciology, 31*, 275-280. doi:10.3189/172756400781820363, 2000

O'Connor, J. M., J. B. McQuitty, and P. C. Clark. "Heat and moisture loads in three commercial broiler breeder barns." *Canadian Agricultural Engineering* 30, no. 2: 267-271. PDF,1987.

Ryan, J. C., Smith, L. C., Wu, M., Cooley, S. W., Miège, C., Montgomery, L. N., et al (2020). Evaluation of Cloudsat's cloud-profiling radar for mapping snowfall rates across the Greenland Ice Sheet. *Journal of Geophysical Research: Atmospheres*, 125, e2019JD031411. https://doi.org/10.1029/2019JD031411